# THINK-ON-GRAPH: DEEP AND RESPONSIBLE REASONING OF LARGE LANGUAGE MODEL ON KNOWLEDGE GRAPH

**Jiashuo Sun**[21*†] **Chengjin Xu**[1*] **Lumingyuan Tang**[31*†] **Saizhuo Wang**[41*]
**Chen Lin**[2] **Yeyun Gong**[6] **Lionel M. Ni**[5] **Heung-Yeung Shum**[14] **Jian Guo**[15‡]
[1]IDEA Research, International Digital Economy Academy
[2]Xiamen University
[3]University of Southern California
[4]The Hong Kong University of Science and Technology
[5]The Hong Kong University of Science and Technology (Guangzhou)
[6]Microsoft Research Asia

## ABSTRACT

Although large language models (LLMs) have achieved significant success in various tasks, they often struggle with hallucination problems, especially in scenarios requiring deep and responsible reasoning. These issues could be partially addressed by introducing external knowledge graphs (KG) in LLM reasoning. In this paper, we propose a new LLM-KG integrating paradigm "LLM $\otimes$ KG" which treats the LLM as an agent to interactively explore related entities and relations on KGs and perform reasoning based on the retrieved knowledge. We further implement this paradigm by introducing a new approach called Think-on-Graph (ToG), in which the LLM agent iteratively executes beam search on KG, discovers the most promising reasoning paths, and returns the most likely reasoning results. We use a number of well-designed experiments to examine and illustrate the following advantages of ToG: 1) compared with LLMs, ToG has better deep reasoning power; 2) ToG has the ability of knowledge traceability and knowledge correctability by leveraging LLMs reasoning and expert feedback; 3) ToG provides a flexible plug-and-play framework for different LLMs, KGs and prompting strategies without any additional training cost; 4) the performance of ToG with small LLM models could exceed large LLM such as GPT-4 in certain scenarios and this reduces the cost of LLM deployment and application. As a training-free method with lower computational cost and better generality, ToG achieves overall SOTA in 6 out of 9 datasets where most previous SOTAs rely on additional training. Our code is publicly available at `https://github.com/IDEA-FinAI/ToG`.

## 1 INTRODUCTION

Large language models (LLMs) (Ouyang et al., 2022; OpenAI, 2023; Thoppilan et al., 2022; Brown et al., 2020a; Chowdhery et al., 2022; Touvron et al., 2023) have demonstrated remarkable performance across various natural language processing tasks. These models capitalize on pre-training techniques applied to vast text corpora to generate responses that are coherent and contextually appropriate. Despite their impressive performance, LLMs have substantial limitations when facing complex knowledge reasoning tasks (Petroni et al., 2021; Talmor et al., 2019; Talmor & Berant, 2018; Zhang et al., 2023) that require deep and responsible reasoning. Firstly, LLMs usually fail to provide accurate answers to questions requiring specialized knowledge beyond what was included in the pre-training phase (out-of-date knowledge in Figure 1a), or to questions requiring long logic chain and multi-hop knowledge reasoning. Secondly, LLMs lack responsibility, explainability and transparency,

---

[*]Equal contribution.
[†]Work done during internship at IDEA Research.
[‡]Corresponding author.

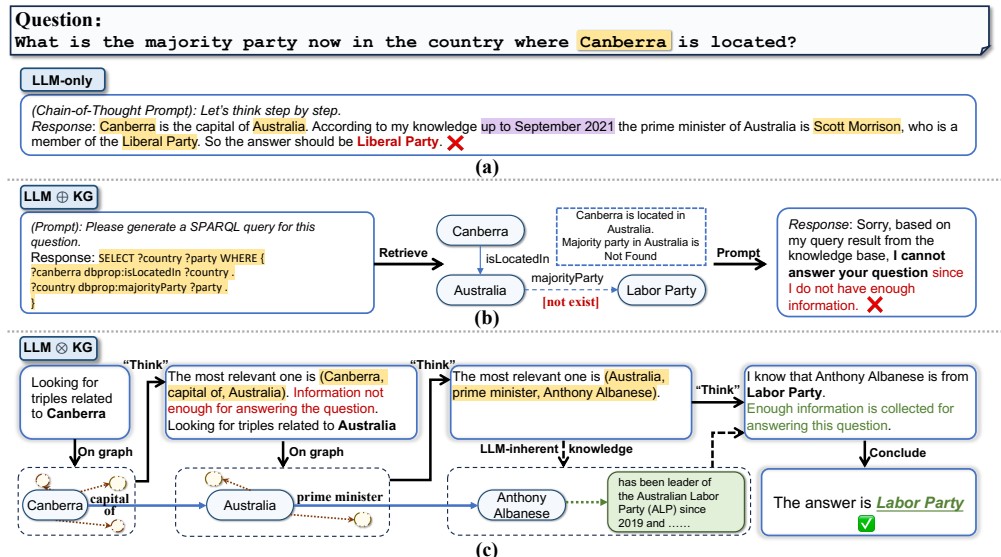

Figure 1: Representative workflow of three LLM reasoning paradigms: (a) LLM-only (e.g., Chain-of-Thought prompting), (b) LLM ⊕ KG (e.g., KBQA via LLM-generated SPARQL query), (c) LLM ⊗ KG (e.g., Think-on-Graph).

raising concerns about the risk of hallucinations or toxic texts. Thirdly, the training process for LLMs is often expensive and time-consuming, making it challenging to keep their knowledge up to date.

Recognizing these challenges, a natural and promising solution is to incorporate external knowledge such as knowledge graphs (KGs) to help improve LLM reasoning. KGs offer structured, explicit, and editable representations of knowledge, presenting a complementary strategy to mitigate the limitations of LLMs (Pan et al., 2023). Researchers (Li et al., 2023b; Xie et al., 2022; Baek et al., 2023b; Yang et al., 2023; Wang et al., 2023a; Jiang et al., 2023) have explored the usage of KGs as external knowledge sources to mitigate hallucination in LLMs. These approaches follow a routine: retrieve information from KGs, augment the prompt accordingly, and feed the increased prompt into LLMs (as illustrated in Figure 1b). In this paper, we refer to this paradigm as "LLM⊕KG". Although aiming to integrate the power of LLM and KG, in this paradigm, LLM plays the role of translator which transfers input questions to machine-understandable command for KG searching and reasoning, but it does not participate in the graph reasoning process directly. Unfortunately, the loose-coupling LLM ⊕ KG paradigm has its own limitations, and its success depends heavily on the completeness and high quality of KG. In Figure 1b, for example, although LLM successfully identified necessary relation types required to answer the question, the absence of the relation "majority party" leads to a failure in retrieving the correct answer.

Building upon these considerations, we propose a new tight-coupling "LLM ⊗ KG" paradigm where KGs and LLMs work in tandem, complementing each other's capabilities in each step of graph reasoning. Figure 1c provides an example illustrating the advantage of LLM ⊗ KG. In this example, the missing relation "majority party" resulting in the failure in Figure 1b can be complemented by a reference triple (**Australia**, **prime minister**, **Anthony Albanese**) discovered by the LLM agent with dynamic reasoning ability (Yao et al., 2022), as well as the political party membership of **Anthony Albanese** coming from LLM's inherent knowledge. In this way, the LLM succeeds in generating the correct answer with reliable knowledge retrieved from KGs. As an implementation of this paradigm, we propose an algorithmic framework "Think-on-Graph" (meaning: LLMs "Think" along the reasoning paths "on" knowledge "graph" step-by-step, abbreviated as ToG below), for deep, responsible, and efficient LLM reasoning. Using the beam search algorithm (Jurafsky & Martin, 2009) in KG/LLM reasoning (Atif et al., 2023; Sun et al., 2023a; Xie et al., 2023; Liu et al., 2024), ToG allows LLM to dynamically explore a number of reasoning paths in KG and make decisions accordingly. Given an input question, ToG first identifies initial entities and then iteratively calls the LLM to retrieve relevant triples from KGs through exploration (looking for relevant triples in KG via "on graph" step) and reasoning (deciding on the most relevant triples via "think" step) until adequate

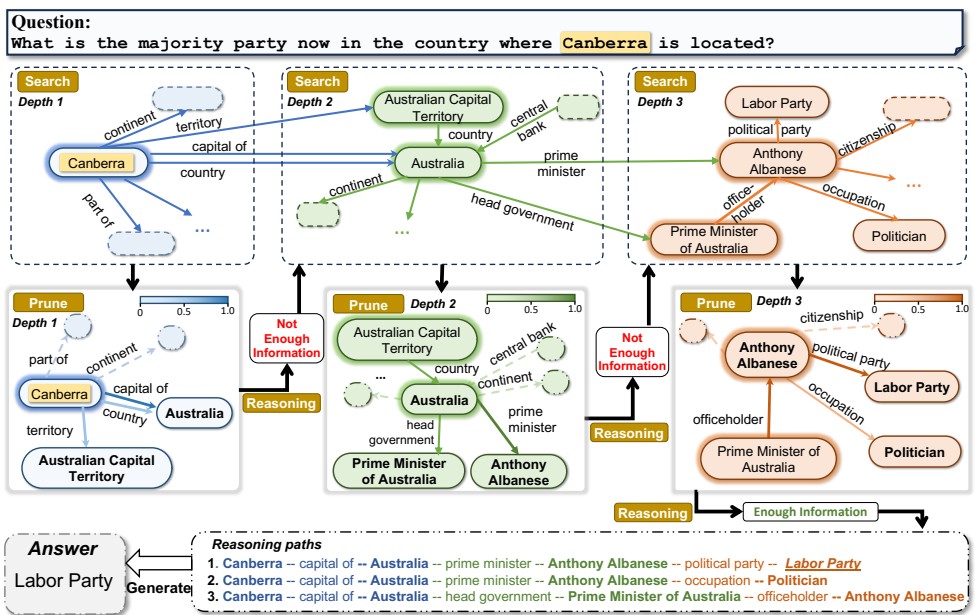

Figure 2: An example workflow of ToG. The glowing entities are the central entities where the search starts at each iteration (depth), and the entities with **boldface** are the selected central entities for the next iteration after pruning. At each pruning step, the darkness of the edges represents the ranking score given by LLM, and the dashed lines indicate relations that have been pruned due to low evaluation scores.

information through the top-N reasoning paths in beam search is gathered to answer the question (judged by LLMs in "Think" step) or the predefined maximum search depth is reached.

The advantage of ToG can be abbreviated as (1) **Deep reasoning:** ToG extracts diverse and multi-hop reasoning paths from KGs as the basis for LLM reasoning, enhancing LLMs' deep reasoning capabilities for knowledge-intensive tasks. (2) **Responsible reasoning:** Explicit, editable reasoning paths improve the explainability of the reasoning process of LLMs, and enable the tracing and correction of the provenances of models' outputs. (3) **Flexibility and efficiency:** a) ToG is a plug-and-play framework that can be applied to a variety of LLMs and KGs seamlessly. b) Under ToG framework, knowledge can be updated frequently via KG instead of LLM whose knowledge-update is expensive and slow. c) ToG enhances the reasoning ability of small LLMs (e.g., LLAMA2-70B) to be competitive with big LLMs (e.g., GPT-4).

## 2 METHODS

ToG implements the "LLM ⊗ KG" paradigm by asking LLM to perform beam search on knowledge graph. Specifically, it prompts the LLM to iteratively explore multiple possible reasoning paths on KGs until the LLM determines that the question can be answered based on the current reasoning paths. ToG constantly updates and maintains top-$N$ reasoning paths $P = \{p_1, p_2, \ldots, p_N\}$ for the question $x$ after each iteration, where $N$ denotes the width of beam search. The entire inference process of ToG contains the following 3 phases: initialization, exploration, and reasoning.

### 2.1 THINK-ON-GRAPH

#### 2.1.1 INITIALIZATION OF GRAPH SEARCH

Given a question, ToG leverages the underlying LLM to localize the initial entity of the reasoning paths on knowledge graph. This phase can be regarded as the initialization of the top-$N$ reasoning paths $P$. ToG first prompts LLMs to automatically extract the topic entities in question and gets the

top-$N$ topic entities $E^0 = \{e_1^0, e_2^0, ..., e_N^0\}$ to the question. Note that the number of topic entities might possibly be less than $N$.

### 2.1.2 EXPLORATION

At the beginning of the $D$-th iteration, each path $p_n$ consists of $D-1$ triples, i.e., $p_n = \{(e_{s,n}^d, r_{j,n}^d, e_{o,n}^d)\}_{d=1}^{D-1}$, where $e_{s,n}^d$ and $e_{o,n}^d$ denote subject and object entities, $r_{j,n}^d$ is a specific relation between them, $(e_{s,n}^d, r_{j,n}^d, e_{o,n}^d)$ and $(e_{s,n}^{d+1}, r_{j,n}^{d+1}, e_{o,n}^{d+1})$ are connected to each other. The sets of the tail entities and relations in $P$ are denoted as $E^{D-1} = \{e_1^{D-1}, e_2^{D-1}, ..., e_N^{D-1}\}$ and $R^{D-1} = \{r_1^{D-1}, r_2^{D-1}, ..., r_N^{D-1}\}$, respectively.

The exploration phase in the $D$-th iteration aims to exploit the LLM to identify the most relevant top-$N$ entities $E^D$ from the neighboring entities of the current top-$N$ entity set $E^{D-1}$ based on the question $x$ and extend the top-$N$ reasoning paths $P$ with $E^D$. To address the complexity of handling numerous neighboring entities with the LLM, we implement a two-step exploration strategy: first, exploring significant relations, and then using selected relations to guide entity exploration.

**Relation Exploration**   Relation exploration is a beam search process with the depth of 1 and the width of $N$ from $E^{D-1}$ to $R^D$. The whole process can be decomposed into two steps: `Search` and `Prune`. The LLM serves as an agent to automatically complete this process.

- **Search** At the beginning of the $D$-th iteration, the relation exploration phase first searches out relations $R_{cand,n}^D$ linked to the tail entity $e_n^{D-1}$ for each reasoning path $p_n$. These relations are aggregated into $R_{cand}^D$. In the case of Figure 2, $E^1 = \{\textbf{Canberra}\}$ and $R_{cand}^1$ denotes the set of all relations linked to **Canberra** inwards or outwards. Notably, the `Search` procedure can be easily completed by executing two simple pre-defined formal queries shown in Appendix E.1 and E.2, which makes ToG adapt well to different KGs **without any training cost**.

- **Prune** Once we have obtained the candidate relation sets $R_{cand}^D$ and the expanded candidate reasoning paths $P_{cand}$ from the relation search, we can utilize the LLM to select out new top-$N$ reasoning paths $P$ ending with the tail relations $R^D$ from $P_{cand}$ based on the literal information of the question $x$ and the candidate relations $R_{cand}^D$. The prompt used here can be found in Appendix E.3.1. As shown in Figure 2, the LLM selects top-3 relations {**capital of**, **country**, **territory**} out from all relations linked to the entity **Canberra** in the first iteration. Since **Canberra** is the only topic entity, the top-3 candidate reasoning paths are updated as {(**Canberra, capital of**), (**Canberra, country**),(**Canberra, territory**)}.

**Entity Exploration**   Similar to relationship exploration, entity exploration is also a beam search process performed by the LLM from $R^D$ to $E^D$, and consists of two steps, `Search` and `Prune`.

- **Search** Once we have obtained new top-$N$ reasoning paths $P$ and the set of new tail relations $R^D$ from relation exploration, for each relation path $p_n \in P$, we can explore a candidate entity set $E_{cand,n}^D$ by querying $(e_n^{D-1}, r_n^D, ?)$ or $(?, r_n^D, e_n^{D-1})$, where $e_n^{D-1}, r_n$ denote the tail entity and relation of $p_n$. We can aggregate $\{E_{cand,1}^D, E_{cand,2}^D, ..., E_{cand,N}^D\}$ into $E_{cand}^D$ and expand top-$N$ reasoning paths $P$ to $P_{cand}$ with the tail entities $E_{cand}^D$. For the shown case, $E_{cand}^1$ can be represented as {**Australia**, **Australia**, **Australian Capital Territory**}.

- **Prune** Since the entities in each candidate set $E_{cand}^D$ is expressed in natural language, we can leverage the LLM to select new top-$N$ reasoning paths $P$ ending with the tail entities $E^D$ out from $P_{cand}$. The prompt used here can be found in Appendix E.3.2. As shown in Figure 2, **Australia** and **Australian Capital Territory** are scored as 1 since the relations **capital of**, **country** and **territory** are only linked to one tail entity respectively, and the current reasoning paths $p$ are updated as {(**Canberra, capital of, Australia**), (**Canberra, country, Australia**), (**Canberra, territory, Australian Capital Territory**)}.

After executing the two explorations described above, we reconstruct new top-$N$ reasoning paths $P$ where the length of each path increases by 1. Each prune step requires at most $N$ LLM calls.

### 2.1.3 REASONING

Upon obtaining the current reasoning path $P$ through the exploration process, we prompt the LLM to evaluate whether the current reasoning paths are adequate for generating the answer. If the evaluation yields a positive result, we prompt the LLM to generate the answer using the reasoning paths with the query as inputs as illustrated in Figure 2. The prompt used for evaluation and generation can be found in Appendix E.3.3 and E.3.4. Conversely, if the evaluation yields a negative result, we repeat the `Exploration` and `Reasoning` steps until the evaluation is positive or reaches the maximum search depth $D_{max}$. If the algorithm has not yet concluded, it signifies that even upon reaching the $D_{max}$, ToG remains unable to explore the reasoning paths to resolve the question. In such a scenario, ToG generates the answer exclusively based on the inherent knowledge in the LLM. The whole inference process of ToG contains $D$ exploration phases and $D$ evaluation steps as well as a generation step, which needs at most $2ND + D + 1$ calls to the LLM.

## 2.2 RELATION-BASED THINK-ON-GRAPH

Previous KBQA methods, particularly based on semantic parsing, have predominantly relied on relation information in questions to generate formal queries (Lan et al., 2022). Inspired by this, we propose relation-based ToG (ToG-R) that explores the top-$N$ relation chains $\{p_n = (e_n^0, r_n^1, r_n^2, ..., r_n^D)\}_{n=1}^N$ starting with the topic entities $\{e_n^0\}_{n=1}^N$ instead of triple-based reasoning paths. ToG-R sequentially performs relation search, relation prune and entity search in each iteration, which is the same as ToG. Then ToG-R performs the reasoning step based on all candidate reasoning paths ending with $E_{cand}^D$ obtained by entity search. If the LLM determines that the retrieved candidate reasoning paths do not contain enough information for the LLM to answer the question, we randomly sample N entities from the candidate entities $E_{cand}^D$ and continue to the next iteration. Assuming that entities in each entity set $E_{cand,n}^D$ probably belong to the same entity class and have similar neighboring relations, the results of pruning the entity set $\{E_{cand,n}^D\}_{n=1}^N$ might have little impact on the following relation exploration. Thus, we use the random beam search instead of the LLM-constrained beam search in ToG for entity prune, referred to as **random prune**. Algorithm 1 and 2 show the implementation details of the ToG and ToG-R. ToG-R needs at most $ND + D + 1$ calls to the LLM.

Compared to ToG, ToG-R offers two key benefits: 1) It eliminates the need for the process of pruning entities using the LLM, thereby reducing the overall cost and reasoning time. 2) ToG-R primarily emphasizes the literal information of relations, mitigating the risk of misguided reasoning when the literal information of intermediate entities is missing or unfamiliar to the LLM.

## 3 EXPERIMENTS

### 3.1 EXPERIMENTAL DESIGN

#### 3.1.1 DATASETS AND EVALUATION METRICS

In order to test ToG's ability on multi-hop knowledge-intensive reasoning tasks, we evaluate ToG on five KBQA datasets (4 Multi-hop and 1 Single-hop): CWQ (Talmor & Berant, 2018), WebQSP (Yih et al., 2016), GrailQA (Gu et al., 2021), QALD10-en (Perevalov et al., 2022), Simple Questions (Bordes et al., 2015). Moreover, in order to examine ToG on more generic tasks, we also prepare one open-domain QA dataset: WebQuestions (Berant et al., 2013); two slot filling datasets: T-REx (ElSahar et al., 2018) and Zero-Shot RE (Petroni et al., 2021); and one fact-checking dataset: Creak (Onoe et al., 2021). Note that, for two big datasets GrailQA and Simple Questions, we only randomly selected 1,000 samples each for testing in order to save computational cost. For all datasets, exact match accuracy (Hits@1) is used as our evaluation metric following previous works (Li et al., 2023b; Baek et al., 2023b; Jiang et al., 2023; Li et al., 2023a).

#### 3.1.2 METHODS SELECTED FOR COMPARISON

We compare with standard prompting (IO prompt) (Brown et al., 2020b), Chain-of-Thought prompting (CoT prompt) (Wei et al., 2022), and Self-Consistency (Wang et al., 2023c) with 6 in-context exemplars and "step-by-step" reasoning chains. Moreover, for each dataset, we pick previous state-of-the-art (SOTA) works for comparison. We notice that fine-tuning methods trained specifically on

| Method | Multi-Hop KBQA | | | | Single-Hop KBQA | Open-Domain QA | Slot Filling | | Fact Checking |
|---|---|---|---|---|---|---|---|---|---|
| | CWQ | WebQSP | GrailQA | QALD10-en | Simple Questions | WebQuestions | T-REx | Zero-Shot RE | Creak |
| *Without external knowledge* | | | | | | | | | |
| IO prompt w/ChatGPT | 37.6 | 63.3 | 29.4 | 42.0 | 20.0 | 48.7 | 33.6 | 27.7 | 89.7 |
| CoT w/ChatGPT | 38.8 | 62.2 | 28.1 | 42.9 | 20.3 | 48.5 | 32.0 | 28.8 | 90.1 |
| SC w/ChatGPT | 45.4 | 61.1 | 29.6 | 45.3 | 18.9 | 50.3 | 41.8 | 45.4 | 90.8 |
| *With external knowledge* | | | | | | | | | |
| Prior FT SOTA | 70.4$^\alpha$ | 82.1$^\beta$ | 75.4$^\gamma$ | 45.4$^\delta$ | 85.8$^\epsilon$ | 56.3$^\zeta$ | 87.7$^\eta$ | 74.6$^\theta$ | 88.2$^\iota$ |
| Prior Prompting SOTA | - | 74.4$^\kappa$ | 53.2$^\kappa$ | - | - | - | - | - | - |
| ToG-R (Ours) w/ChatGPT | 58.9 | 75.8 | 56.4 | 48.6 | 45.4 | 53.2 | 75.3 | 86.5 | 93.8 |
| ToG (Ours) w/ChatGPT | 57.1 | 76.2 | 68.7 | 50.2 | 53.6 | 54.5 | 76.8 | 88.0 | 91.2 |
| ToG-R (Ours) w/GPT-4 | **69.5** | 81.9 | 80.3 | **54.7** | 58.6 | 57.1 | 75.5 | 86.9 | 95.4 |
| ToG (Ours) w/GPT-4 | 67.6 | **82.6** | **81.4** | 53.8 | **66.7** | **57.9** | **77.1** | **88.3** | **95.6** |

Table 1: The ToG results for different datasets. The prior FT (Fine-tuned) and prompting SOTA include the best-known results: $\alpha$: Das et al. (2021); $\beta$: Yu et al. (2023); $\gamma$: Gu et al. (2023); $\delta$: Santana et al. (2022); $\epsilon$: Baek et al. (2023a); $\zeta$: Kedia et al. (2022); $\eta$: Glass et al. (2022); $\theta$: Petroni et al. (2021); $\iota$: Yu et al. (2022); $\kappa$: Li et al. (2023a).

evaluated datasets usually have an advantage by nature over methods based on prompting without training, but sacrificing the flexibility and generalization on other data. For a fair play, therefore, we compare with previous SOTA among all prompting-based methods and previous SOTA among all methods respectively. Note that the paper Tan et al. (2023) is not involved in comparison because its results are not based on standard exact match and thus incomparable.

### 3.1.3 EXPERIMENT DETAILS

Given the plug-and-play convenience of ToG, we try three LLMs in experiments: ChatGPT, GPT-4 and Llama-2. We use OpenAI API to call ChatGPT (GPT-3.5-turbo) and GPT-4[1]. Llama-2-70B-Chat (Touvron et al., 2023) runs with 8 A100-40G without quantization, where the temperature parameter is set to 0.4 for exploration process (increasing diversity) and set to 0 for reasoning process (guaranteeing reproducibility). The maximum token length for the generation is set to 256. In all experiments, we set both width $N$ and depth $D_{max}$ to 3 for beam search. Freebase (Bollacker et al., 2008) is used as KG for CWQ, WebQSP, GrailQA, Simple Questions, and Webquestions, and Wikidata (Vrandečić & Krötzsch, 2014) is used as KG for QALD10-en, T-REx, Zero-Shot RE and Creak. We use 5 shots in ToG-reasoning prompts for all the datasets.

### 3.2 MAIN RESULTS

### 3.2.1 COMPARISON TO OTHER METHODS

Since CoT uses external KG to enhance LLM, we first compare it with those methods leveraging external knowledge as well. As we can see in Figure 1, even if ToG is a training-free prompting-based method and has natural disadvantage in comparison with those fine-tuning methods trained with data for evaluation, ToG

| Method | CWQ | WebQSP |
|---|---|---|
| *Fine-tuned* | | |
| NSM (He et al., 2021) | 53.9 | 74.3 |
| CBR-KBQA (Das et al., 2021) | 67.1 | - |
| TIARA (Shu et al., 2022) | - | 75.2 |
| DeCAF (Yu et al., 2023) | 70.4 | 82.1 |
| *Prompting* | | |
| KD-CoT (Wang et al., 2023b) | 50.5 | 73.7 |
| StructGPT (Jiang et al., 2023) | - | 72.6 |
| KB-BINDER (Li et al., 2023a) | - | 74.4 |
| *LLama2-70B-Chat* | | |
| CoT | 39.1 | 57.4 |
| ToG-R | **57.6** | **68.9** |
| ToG | 53.6 | 63.7 |
| Gain | (+18.5) | (+11.5) |
| *ChatGPT* | | |
| CoT | 38.8 | 62.2 |
| ToG-R | 57.1 | 75.8 |
| ToG | **58.9** | **76.2** |
| Gain | (+20.1) | (+14.0) |
| *GPT-4* | | |
| CoT | 46.0 | 67.3 |
| ToG-R | 67.6 | 81.9 |
| ToG | **69.5** | **82.6** |
| Gain | (+23.5) | (+15.3) |

Table 2: Performances of ToG using different backbone models on CWQ and WebQSP.

with GPT-4 still achieves new SOTA performance in 6 out of 9 datasets, including WebQSP, GrailQA,

---

[1]GPT-3.5-turbo and GPT-4 is both from https://openai.com/

QALD10-en, WebQuestions, Zero-Shot RE and Creak. Even for some dataset without SOTA, e.g., CWQ, the performance of CoT has already been close to SOTA (69.5% v.s. 70.4%). If comparing with all promoting-based methods, both ToG with GPT-4 and its weaker version ToG with ChatGPT can win the competition in all datasets. In particular, the improvement of 1.6% on open-domain QA dataset WebQuestions demonstrates the ToG's generality on open-domain QA tasks. We also notice that the performance of ToG on single-hop KBQA dataset is not as good as its performance on other datasets. These results indicate that ToG is more effective on multi-hop datasets in general, which supports our argument that ToG enhances the deep reasoning capability of LLMs.

We also see from Figure 1 that, compared with those methods without leveraging external knowledge (e.g, IO, CoT and SC prompting methods), the advantage of ToG is more significant. For example, the performance improves 51.8% and 42.9% on GrailQA and Zero-Shot RE, respectively. It turns out that benefits from external KG can not be ignored in reasoning.

ToG outperforms ToG-R on most datasets since the triple-based reasoning paths provide additional intermediate entity information compared to the relation chains retrieved by ToG-R. More detailed analysis of the answers generated by ToG can be checked in Appendix B.2. And the results of previous methods on each dataset are reported in Appendix C for better comparison,

### 3.2.2 PERFORMANCES WITH DIFFERENT BACKBONE MODELS

Given ToG's flexibility of plug-and-play, we evaluate how different backbone models affect its performance on two datasets CWQ and WebQSP. Table 2 shows that, as we expected, the performance of CoT improves with the size (also reflecting partially the reasoning ability) of backbone models (GPT-4 > ChatGPT > Llama-2). Furthermore, we see that, the larger the backbone model, the larger the gap between CoT and ToG (the gain increases from 18.5% for Llama-2 to 23.5% for GPT-4 on CWQ, and from 11.5% for Llama-2 to 15.3% for GPT-4 on WebQSP), and this indicates more potential of KG can be mined using a more powerful LLM.

In addition, even if using the smallest model Llama-2 (70B parameters), ToG outperforms CoT with GPT-4. This implies a much cheaper technical route for LLM deployment and application, i.e., TOG with cheap small LLM may be a candidate for substituting expensive big LLM, especially in vertical scenarios that external KGs can cover.

### 3.2.3 ABLATION STUDY

We perform various ablation studies to understand the importance of different factors in ToG. We conduct our ablation studies on two subsets of the test sets of CWQ and WebQSP, each of which contains 1,000 randomly sampled questions.

**Do search depth and width matter for ToG?** To explore the influence of the search depth $D_{max}$ and the beam width $N$ on ToG's performance, we conduct experiments under settings with depths ranging from 1 to 4 and widths from 1 to 4. As shown in Figure 3, ToG's performance improves with the search depth and width. This also implies that ToG's performance could potentially be improved with the increment of the exploration depth and breadth. However, considering the computational cost (which increases linearly with the depth), we set both the depth and width to 3 as the default experimental setting. On the other hand, the performance growth diminishes when the depth exceeds 3. This is mainly because only a small part of questions have the reasoning depths (based on the number of relations in SPARQL, as seen in Figure 12 in the Appendix) of greater than 3.

| Method | CWQ | WebQSP |
|---|---|---|
| **CoT** | 37.6 | 62.0 |
| **ToG** | | |
| w/ Freebase | 58.8 | 76.2 |
| w/ WikiData | 54.9 | 68.6 |
| **ToG-R** | | |
| w/ Freebase | 59.2 | 75.1 |
| w/ WikiData | 51.9 | 66.7 |

Table 3: Performances of ToG using different source KGs on CWQ and WebQSP.

**Do different KGs affect ToG's performance?** One of the main advantages of ToG is its plug-and-play capabilities. As shown in Table 3, ToG achieves significant improvements with different source KGs on CWQ and WebQSP, compared to CoT. On the other hand, different source KGs might have

Figure 3: Performances of ToG with different search depths and widths.

different effects on the performance of ToG. Notably, Freebase brings more significant improvements on CWQ and WebQSP than Wikidata, since both datasets are constructed upon Freebase. Moreover, in a very large KG like Wikidata, the searching and pruning processes are relatively challenging.

**How do different prompt designs affect ToG?** We perform additional experiments to determine which types of prompt representations can work well for our approach. The results are presented in Table 4. "Triples" denotes using triple formats as prompts to represent multiple paths, such as "(Canberra, capital of, Australia), (Australia, prime minister, Anthony Albanese)". "Sequences" refers to the utilization of a sequence format, as illustrated in Figure 2. "Sentences" involves converting the triples into natural language sentences. For example, "(Canberra, capital of, Australia)" can be converted to "The capital of Canberra is Australia." The result shows that the utilization of triple-based representations for the reasoning paths yields the highest degree of efficiency and superior performance. Conversely, when considering ToG-R, each reasoning path

| Method | CWQ | WebQSP |
|---|---|---|
| **ToG** | | |
| w/ Triples | 58.8 | 76.2 |
| w/ Sequences | 57.2 | 73.2 |
| w/ Sentences | 58.6 | 73 |
| **ToG-R** | | |
| w/ Sequences | 59.2 | 75.1 |
| w/ Sentences | 50.1 | 67.3 |

Table 4: Performances of ToG using different prompting designs.

is a relation chain starting from a topic entity, rendering it incompatible with the triple-based prompt representation. Consequently, the transformation of ToG-R into the natural language form results in excessively lengthy prompts, thereby leading to a notable deterioration in performance.

**Comparing the affects from different pruning tools.** Other than the LLM, lightweight models that can measure text similarity like BM25 and SentenceBERT, can be employed as pruning tools in the exploration phase. We can select top-$N$ entities and relations based on their literal similarities with the question. We investigate the impacts of different pruning tools on the performance of the ToG, as demonstrated in Table 5. The replacement of the LLM with either BM25 or SentenceBERT results in the significant performance degradation of our approach. Concretely, the results on CWQ drop on average by 8.4%, and the results on WebQSP drop on average by 15.1%. The results show that the LLMs perform best as a pruning

| Method | CWQ | WebQSP |
|---|---|---|
| **ToG** | | |
| w/BM25 | 51.4 | 58.7 |
| w/SentenceBERT | 51.7 | 66.3 |
| w/ChatGPT | 58.8 | 76.2 |
| **ToG-R** | | |
| w/BM25 | 49.4 | 57.3 |
| w/SentenceBERT | 50.1 | 60.1 |
| w/ChatGPT | 59.2 | 75.1 |

Table 5: Performances of ToG using different pruning tools.

tool in terms of effectiveness. On the other hand, after utilizing the BM25 or SentenceBERT, we only need $D + 1$ calls to the LLM instead of $2ND + D + 1$ as we discuss in Section 2.1.3, which enhances the efficiency of ToG.

We conduct additional ablation studies on the effect of the number of seed exemplars and the difference between ToG and naive beam search on the KG, which can be seen in Appendix B.1.

## 3.3 KNOWLEDGE TRACEABILITY AND CORRECTABILITY IN ToG

The quality of KG is very important for correct reasoning by ToG. An interesting feature of ToG is knowledge traceability and knowledge correctability during LLM reasoning, and it provides a way to improve KG's quality using ToG itself and reduce the cost of KG construction and correction. As illustrated in Figure 4, the explicit reasoning paths of the ToGs can be displayed to users. If potential

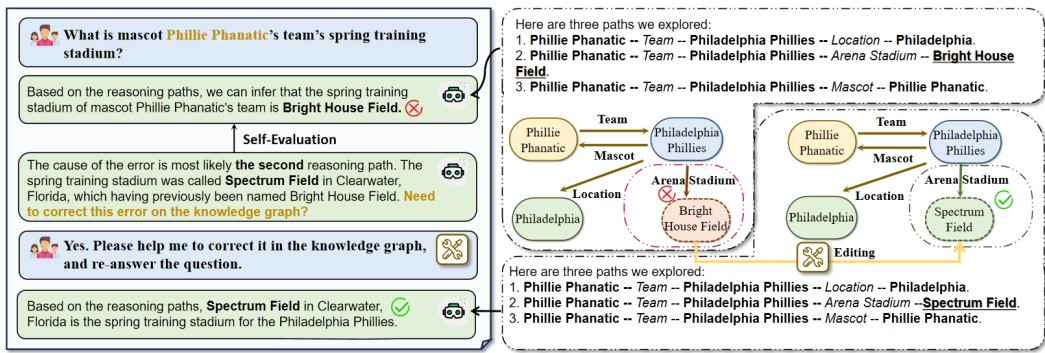

Figure 4: The illustration of knowledge traceability and correctability of ToG.

errors in ToG answers are discovered by human users/experts or other LLMs, ToG has the ability to trace back and examine the reasoning path, find suspicious triples with errors, and correct them.

Take the case in Figure 4 as an example. Given the input question "What is mascot Phillie Phanatic's team's spring training stadium?", ToG outputs the wrong answer "Bright House Field" in the first round. Then ToG traces back all reasoning paths, localizes the cause of the error may come from the second reasoning path (Phillie Phanatic $\xrightarrow{\text{Team}}$ Philadelphia Phillies $\xrightarrow{\text{Arena Stadium}}$ Bright House Field), and analyzes that the error comes from the old name "Sp​ecturm Field" of "Bright House Field" in the outdated triple (*Philadelphia Phillies*, *Arena Stadium*, *Bright House Field*). According to the hints from ToG, the user can ask LLM to correct this error and answer the same question with the correct information. This example reveals that ToG not only enhances LLM with KG, but also improves the quality of KG with LLM, known as knowledge infusion (Moiseev et al., 2022).

## 4 RELATED WORK

**Reasoning with LLM Prompting** Chain-of-Thought (CoT) (Wei et al., 2022) has been shown to be effective in enhancing LLM reasoning. It creates a series of prompt instances according to reasoning logic under a few-shot learning paradigm in order to improve LLM's performance on complex tasks. The thought of CoT has been improved along different dimensions, including Auto-CoT (Zhang et al., 2022), Complex-CoT (Fu et al., 2023), Self-Consistency (Wang et al., 2023c), Iter-CoT (Sun et al., 2023b), ToT (Yao et al., 2023), GoT (Besta et al., 2023) and so on. Given the limitation that all these works only use the knowledge in training data, recent efforts such as ReAct (Yao et al., 2022) attempt to utilize the information from external sources to further improve the reasoning performance.

**KG-enhanced LLM** KG has advantages in dynamic, explicit, and structured knowledge representation and techniques combining LLMs with KGs have been studied. Early studies (Peters et al., 2019; Luo et al., 2024) embed structured knowledge from KGs into the underlying neural networks during the pretraining or fine-tuning process. However, KG embedded in LLM sacrifices its own nature of explainability in knowledge reasoning and efficiency in knowledge updating (Hu et al., 2023).

Recent works instead combine LLMs with KGs by translating relevant structured knowledge from KGs to textual prompts for LLMs. All the methods belong to the LLM ⊕ KG paradigm we defined in the introduction section. On the other hand, Jiang et al. (2023) asks LLM to explore KG and so it can be regarded as a special case of ToG, which belongs to the LLM ⊗ KG paradigms.

## 5 CONCLUSION

We introduce the LLM ⊗ KG paradigm for integrating LLMs and KGs and propose the Think-on-Graph (ToG) framework, which leverages LLM as an agent participating in KG reasoning for better decision-making. Experimental results demonstrate that ToG outperforms existing fine-tuning-based methods and prompting-based methods without additional training cost and mitigates the hallucination issue of LLMs.

## 6 Acknowledgement

We express our sincere gratitude to the esteemed reviewers for their invaluable feedback and constructive comments, which significantly contributed to the improvement and refinement of this paper. Their insightful suggestions and meticulous attention to detail have played a pivotal role in enhancing the quality and clarity of our research work.

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

# A    ALGORITHM FOR ToG

We summarize the comprehensive algorithmic procedure of ToG and ToG-R, as shown in Figure Algorithm 1 and 2.

| **Algorithm 1** ToG | **Algorithm 2** ToG-R |
|---|---|
| **Require:** Input $x$, LLM $\pi$, depth limit $D_{max}$ sample limit $N$.
    Initialize $E^0 \leftarrow$ Extract entities on $x$, $P \leftarrow []$, $M \leftarrow 0$.
    **while** $D \leq D_{max}$ **do**
        $R^D_{cand}, P_{cand} \leftarrow$ Search($x$, $E^{D-1}$, $P$)
        $R^D, P \leftarrow$ Prune($\pi$, $x$, $R^D_{cand}, P_{cand}$)
        $E^D_{cand}, P_{cand} \leftarrow$ Search($x$, $E^{D-1}$, $R^D$, $P$)
        $E^D, P \leftarrow$ Prune($\pi$, $x$, $E^D_{cand}, P_{cand}$)
        **if** Reasoning($\pi$, $x$, $P$) **then**
            Generate($\pi$, $x$, $P$)
            **break**
        **end if**
        Increment $D$ by 1.
    **end while**
    **if** $D > D_{max}$ **then**
        Generate($\pi$, $x$)
    **end if** | **Require:** Input $x$, LLM $\pi$, depth limit $D_{max}$ sample limit $N$.
    Initialize $E^0 \leftarrow$ Extract entities on $x$, $P \leftarrow []$, $M \leftarrow 0$.
    **while** $D \leq D_{max}$ **do**
        $R^D_{cand}, P_{cand} \leftarrow$ Search($x$, $E^{D-1}$, $P$)
        $R^D, P \leftarrow$ Prune($\pi$, $x$, $R^D_{cand}, P_{cand}$)
        $E^D_{cand}, P_{cand} \leftarrow$ Search($x$, $E^{D-1}$, $R^D$, $P$)
        **if** Reasoning($\pi$, $x$, $P$, $E^D_{cand}$) **then**
            Generate($\pi$, $x$, $P$, $E^D_{cand}$)
            **break**
        **end if**
        $E^D, P \leftarrow$ Random_Prune($E^D_{cand}, P_{cand}$)
        Increment $D$ by 1.
    **end while**
    **if** $D > D_{max}$ **then**
        Generate($\pi$, $x$)
    **end if** |

# B    ADDITIONAL ABLATION STUDY AND EXPERIMENT ANALYSIS

In this section, we conduct more experiments for ablation study in addition to Section 3.2.3, and analyze experimental results of ToG in detail.

## B.1    ADDITIONAL ABLATION STUDY

**Sensitivity to the Number of Seed Examplars**    To better understand how sensitive ToG is sensitivity to the number of seed exemplars, we employ sensitivity analysis shown in Figure 5. We conduct zero-shot experiment and select 1-6 examples from the training set as few-shot setting. In the few-shot tests, we randomly chose $M$ of $\{1, 2, 3, 4, 6\}$ exemplars as demonstrations and replicated the experiments three times. As the number of examples in the demonstrations increases, the overall performance also generally improves. However, the performance peaks for ToG and ToG-R differ (with the best performance for ToG at 5-shot and for ToG-R at 4-shot). Moreover, ToG's zero-shot performance outpaces ToG-R. This can be attributed to ToG having fully completely explored paths, ensuring commendable performance even in zero-shot. In contrast, ToG-R omits entities in the path, but its average performance with demonstrations is superior to ToG.

| Search Algorithm | Dataset | EM |
|---|---|---|
| Naive Beam Search | CWQ | 30.1 |
|  | WebQSP | 46.1 |
| TOG-R | CWQ | 59.2 |
|  | WebQSP | 75.1 |
| TOG | CWQ | 58.8 |
|  | WebQSP | 76.2 |

Table 6: The results of Naive Beam Search, ToG methods on CWQ and WebQSP.

**Difference with Naive Beam Search**    ToG is slightly different from the beam search. ToG uses the top-$N$ reasoning paths as evidence while the naive beam search chooses the most plausible path as the only reasoning path. We conduct naive top1-beam search methods for ToG on CWQ and WebQSP. For each depth of the ToG, we choose the reasoning path with the highest plausibility, to evaluate if

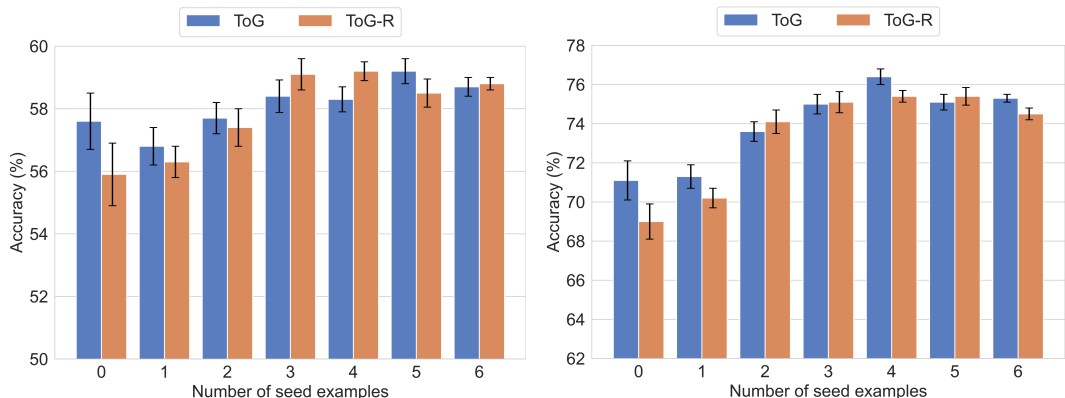

Figure 5: Exemplar sensitivity analysis for CWQ and WebQSP for ToG, where "0" denotes zero-shot and "k" denotes k-shot.

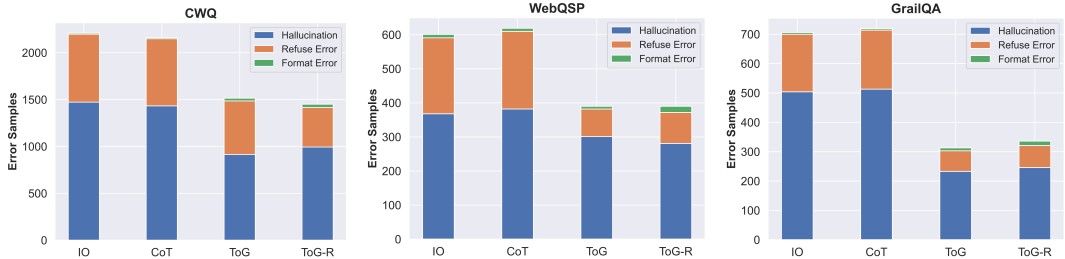

Figure 6: The erroneous instances and categories in the CWQ, WebQSP, and GrailQA of IO, CoT, and ToG.

the current reasoning path is sufficient to answer the questions. The experiment results are shown in Table 6. In naive beam search, the calibration error accumulates along the inference, leading to the instability of the final result. We believe that ToG can partially alleviate this issue by considering the top-$N$ reasoning paths.

## B.2 RESULT ANALYSIS

We conduct a detailed analysis on the answers generated by ToG and ToG-R.

**Error Analysis** We considered three types of errors: (1) Hallucination error, (2) Refuse error [2], and (3) Format error. The distribution is shown in Figure 6. Our approach has significantly reduced the hallucination and refusal to answer error types in IO and CoT. For GrailQA, ToG even reduces these types of errors by 50% and 60%, respectively. Moreover, in ToG's error samples, there are still many instances of hallucination and refusal to answer errors. This is because the current search depth and width are both set to 3. By increasing the search depth and width, these error instances will further decrease (refer to Section 3.2.3). Furthermore, we currently generalize incorrect answers as hallucinations, but there are various categories within hallucinations, which we won't discuss in this paper. Additionally, after applying ToG, there's a slight increase in samples with format errors. This result shows that the explored paths lead to a noticeable increase in the tokens, sometimes even exceeding the maximum output limit. However, the error rate from this issue is negligible (less than 3%).

**Evidence of Answers** We conducted an analysis of the correctly answered samples in three datasets to investigate the evidence for LLM in generating answers as shown in Figure 7. Evidently, a significant portion of the answers are derived from the paths explored by ToG, while roughly 20% rely exclusively on the intrinsic knowledge embedded within LLM's parameters for generating

---

[2]LLM will refuse to answer due to lack of information.

responses. It is worth noting that around 7% of the correctly answered samples require a combination of knowledge from both the explored paths and LLM's inherent knowledge (as elaborated in Appendix Table 21). This distinction sets our approach apart from traditional graph-based search methods, as it does not necessitate the path to encompass the node containing the correct answer entirely. Instead, the explored paths supplement and reference LLM's inherent knowledge. The distribution of answer types for ToG-R is almost indistinguishable from that of ToG, proving the robustness of our approach.

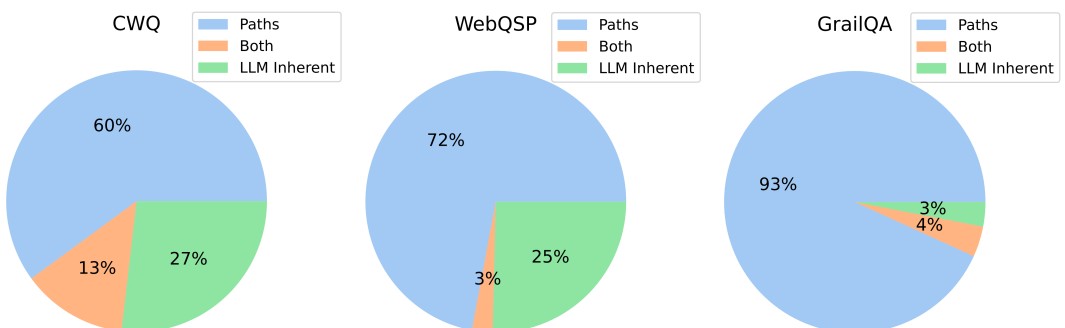

Figure 7: The proportions of ToG's evidence of answers on CWQ, WebQSP, and GrailQA datasets.

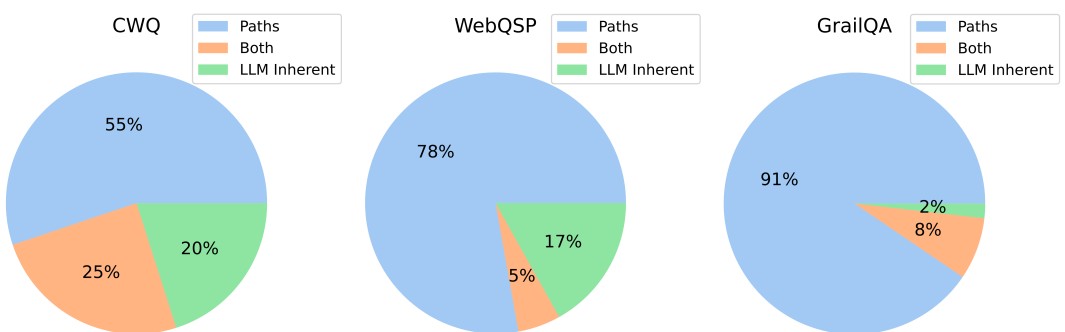

Figure 8: The explored path overlap ratio of ToG-R on CWQ, WebQSP, and GrailQA datasets.

**The Overlap Ratio between the Explored Paths and Ground-truth Paths**  We also conduct an analysis of the correctly answered samples in three datasets to investigate the ratio of overlap between the paths explored by ToG and the ground-truth path in SPARQL. The definition of overlap ratio is the ratio of overlapping paths to the total number of relations in ground-truth SPARQL:

$$\frac{\mathrm{Count}(\mathrm{Rel}(\mathrm{Paths}) \cap \mathrm{Rel}(\mathrm{SPARQL}))}{\mathrm{Count}(\mathrm{Rel}(\mathrm{SPARQL}))}$$

where Rel(*) denotes all the unduplicated relations in the "*" and Count(*) denotes the number of "*"[3]. Figure 9 is a path schematic which takes the case shown in Table 22 for example. It can be observed from Figure 10 that the paths explored by ToG are identical to the golden paths of an average of 30% correct samples, while the paths of an average of 21% correct samples are completely different from the golden path. This indicates that ToG has successfully explored a completely and approximately new path in the knowledge graph space to reach the final answer entity. For ToG-R, the disparity between the two is primarily evident in the CWQ dataset, where the percentage of intervals (25,50] in ToG results is quite significant (nearly 40%), whereas ToG-R results tend to be more evenly distributed as shown in Figure 11. We contend that this discrepancy arises from the disregard of entity, thereby enhancing the diversity of explored relations. This represents a significant application of knowledge graph reasoning in academic research.

---

[3]We approximately calculate the length of a path by counting the number of relations in the ground-truth SPARQL.

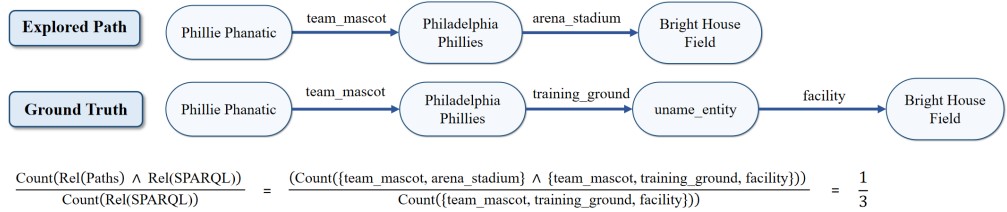

$$\frac{\text{Count}(\text{Rel}(\text{Paths}) \wedge \text{Rel}(\text{SPARQL}))}{\text{Count}(\text{Rel}(\text{SPARQL}))} = \frac{(\text{Count}(\{\text{team\_mascot}, \text{arena\_stadium}\} \wedge \{\text{team\_mascot}, \text{training\_ground}, \text{facility}\}))}{\text{Count}(\{\text{team\_mascot}, \text{training\_ground}, \text{facility}\}))} = \frac{1}{3}$$

Figure 9: Path schematic to calculate overlap.

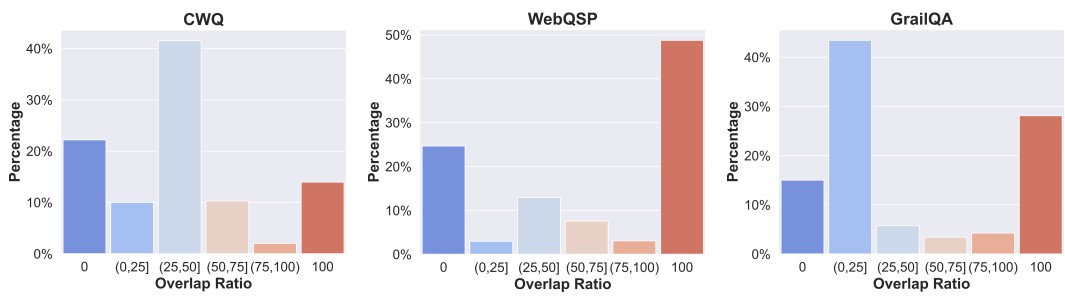

Figure 10: The explored path overlap ratio of ToG on CWQ, WebQSP, and GrailQA datasets.

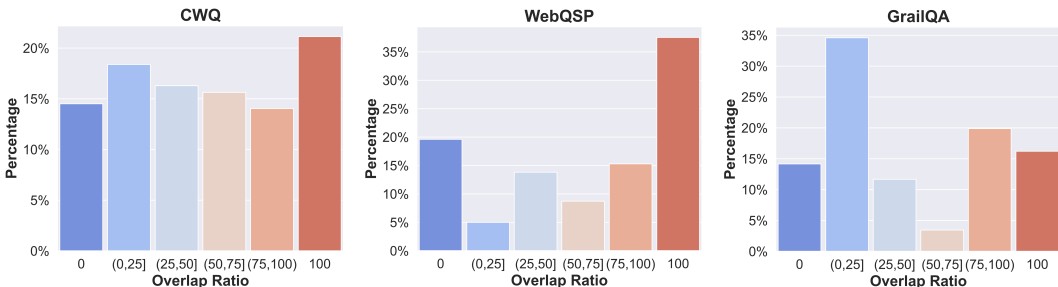

Figure 11: The path overlap ratio of ToG-R on CWQ, WebQSP, and GrailQA datasets.

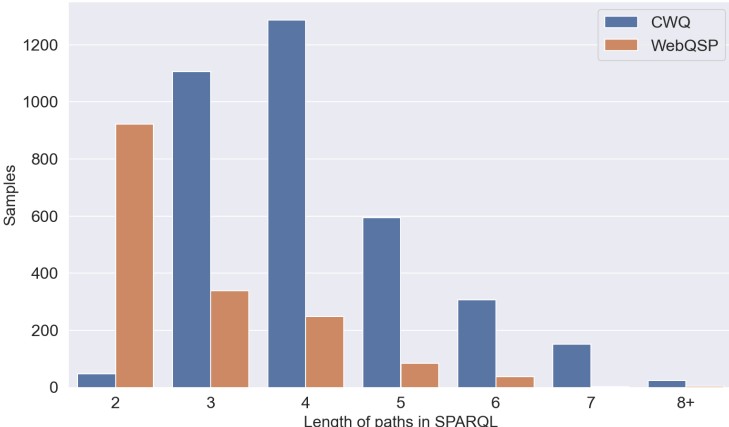

Figure 12: The lengths of the ground-truth SPARQL queries within the CWQ and WebQSP datasets, computed based on relation numbers.

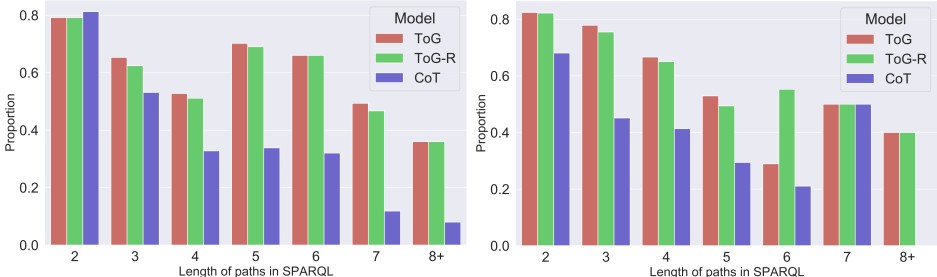

Figure 13: ToG, ToG-R and CoT's performance among CWQ and WebQSP dataset.

**The Reasoning Depth of Questions**   We calculate the reasoning depth of testing questions based on the number of relations within their ground-truth SPARQL queries on CWQ and WebQSP. The counts of questions with different reasoning depths are shown in Figure 12. We analyze the performances of ToG, ToG-R, and CoT on testing questions of both datasets with different reasoning depths. As illustrated in Figure 13, the performances of CoT show roughly decreasing trends on both datasets, with the reasoning depth of testing questions increasing. Conversely, ToG and ToG-R can partially counteract the performance degradation caused by the increment of reasoning depths of questions, especially on CWQ. Generally, the performance difference between ToG and CoT becomes more significant on deeper questions.

### B.2.1   EFFICIENCY OF TOG

There are many solutions to improve efficiency and reduce the computational complexity (proportional to the number of calling LLMs) of ToG from the original $O(ND)$ to $O(D)$, where $D$ is the depth (or equivalently length) of the reasoning path, and $N$ is the width of the beam-search (how many paths are remained in the pool in each iteration).

**Solution 1**   Reducing computational complexity from $O(ND)$ to $O(D)$ by using lightweight model in pruning. The bottleneck of computation is the pruning step, which contributes to $N * D$ times calling, and it is important to optimize it for computational efficiency. A technical route is to replace LLM with small models such as BM25 and Sentence-BERT in the pruning step since the small models are much faster than LLM calling. In this way, we can reduce the number of LLM calling from $2ND + D + 1$ to $D + 1$. When $D$=3, for example, there are only 4 times LLM calling. However, this optimization sacrifices the accuracy due to the weaker scoring model in pruning. For instance, as shown in Table 5 of the manuscript, the performance of ToG on WebQSP drops from 76.2% to 66.3% after replacing ChatGPT with SentenceBERT for pruning. To alleviate the issue of the performance degradation, we can appropriately increase the search width to compensate the loss because increasing search width can improve the chance of the optimal path to be selected in the pool and it doesn't affect the number of LLM calling. To empirically verify this, we increase the search width from 3 to 5 and reevaluate ToG with SentenceBERT as the pruning model on WebQSP. The accuracy rises to from 66.3% to 68.5% and could be further improved with a greater width since the greater width would not cause an increase in the number of LLM calls.

**Solution 2**   Reducing computational complexity from $O(ND)$ to $O(D)$ by unifying the prompts in the same pruning step. Another solution on speeding up the pruning step is to employ the LLM at once to score all components of N candidate sets for obtaining top-N candidates, instead of calling the LLM N times to score N candidate sets separately. Through this solution, either entity pruning step or relation pruning step only need 1 LLM call for each iteration. Thus, the maximum number of LLM calls per question needed for ToG and ToG-R would drop to $2D + D + 1$ and $D + D + 1$.

**Solution 3**   Optimizing pruning step to make the actual calls of LLMs much less than the previously estimated $2ND + D + 1$ and closer to some common prompting methods such as CoT-SC. For ToG and other LLM-based methods, the computational time (cost or complexity) in the inference phase mainly depends on how many times calling LLM. For each question, ToG needs at most $2ND + D + 1$ times. Meanwhile, ToG-R needs at most $ND + D + 1$ times as mentioned in Section 2.

| Dataset | CWQ | WebQSP | GrailQA | QALD10-EN | SimpleQuestion |
|---|---|---|---|---|---|
| Average LLM Calls | 14.3 | 11.2 | 10.4 | 11.4 | 8.7 |

Table 7: Average Number of LLM Calls per Question (Part 1)

| Dataset | WebQuestion | T-REx | Zero-Shot RE | Creak |
|---|---|---|---|---|
| Average LLM Calls | 10.5 | 7.7 | 7.6 | 8.0 |

Table 8: Average Number of LLM Calls per Question (Part 2)

Given the beam search width $N$ and maximal reasoning depth $D$, ToG's initialize the search from the entity mostly aligning with the keyword in question. In each iterative step of the reasoning path, ToG starts from each of the $N$ entities/relations (nodes/edges on knowledge graph) and searches all its neighboring relations/entities. Given the search width $N$, ToG always keep $N$ "most-likely" candidate reasoning paths in the pool, and thus there are always $N$ candidate entity sets $E_{cand,n}^{D}$ and $N$ candidate relation sets $R_{cand,n}^{D}$. Consequently, it needs $N$ LLM calls for entity pruning and $N$ calls for relation pruning, respectively, as well as one additional LLM call for reasoning (evaluating if the information from the current candidate paths are enough or not). We have to point it out that, for each of the $N$ starting entities, all its neighbor entities/relations are NOT scored one by one. On the contrary, all its neighbor entities/relations are "translated" into "one" prompt altogether and are sent to LLM, which output the top-$N$ candidates at one-time. Therefore, each starting entity only calls LLM once for pruning and so $N$ starting entities calls LLM $N$ times in one iterative step. Consequently, there are totally $2ND + D$ times calling after reasoning $D$ steps. In the end, there is an additional calling that "translate" the final path to user-understandable language and answer the user. Therefore, ToG requires $2ND + D + 1$ LLM calls in total. Since most questions can be answered within 3 hops (means depth of reasoning path is 3), and the performance is usually good enough when the search width $N$=3 as we tested in Figure 3, the total number of LLM calling is $2 \times 3 \times 3 + 3 + 1 = 22$. So the computational time is about 21 times longer than that of LLM-only. With a similar performance to ToG, its variant ToG-R only calls LLM for $ND + D + 1$ times by using random entity pruning instead of LLM-based entity pruning, saving nearly half of computational time.

$2ND + D + 1$ is the maximal computational complexity. In most cases, ToG does not need $2ND + D + 1$ LLM calls for a question because the whole reasoning process might be early stopped before the maximum reasoning depth D is reached if LLM determines enough information has been retrieved. Likewise, ToG-R does not really need $ND + D + 1$ LLM calls in most cases. As an illustration, Table 7 and Table 8 show the average numbers of LLM calls per question needed by ToG on different datasets. It can be seen that in the four multi-hop KBQA datasets, the average numbers of LLM calls (ranging from 10 to 15) are significantly smaller than 22, which is the theoretical maximum number of LLM calls calculated from $2ND + D + 1$ when $N$=3 and $D$=3. We can also see that this AVERAGE number gets even smaller (< 10) for single-hop reasoning datasets, such as SimpleQuestion and T-REx.

## C  DATASET

The statistics of the datasets used in this paper are shown in Table 9. We also provide a detailed result table for each dataset, shown in Table 10 to Table 18, illustrating the enhancements of ToG compared to the previous fine-tuning-based and prompting-based relevant works. For QALD10-en, WebQuestions, Zero-Shot RE, and Creak, ChatGPT-based ToG reached a new state-of-the-art. Furthermore, GPT-4-based ToG exceeded the fine-tuning-based approaches on almost all Multi-Hop KBQA datasets, where on CWQ, ToG is close to the state-of-the-art (69.5%).

| Dataset | Answer Format | Train | Test | Licence |
|---|---|---|---|---|
| ComplexWebQuestions | Entity | 27,734 | 3,531 | - |
| WebQSP | Entity/Number | 3,098 | 1,639 | CC License |
| GrailQA* | Entity/Number | 44,337 | 1,000 | - |
| QALD-10 | Entity/Number | - | 333 | MIT License |
| Simple Quesiton* | Entity/Number | 14,894 | 1,000 | CC License |
| WebQuestions | Entity/Number | 3,778 | 2,032 | - |
| T-REx | Entity | 2,284,168 | 5,000 | MIT License |
| Zero-Shot RE | Entity | 147,909 | 3,724 | MIT License |
| Creak | Bool | 10,176 | 1,371 | MIT License |

Table 9: The statistics of the datasets used in this paper. * denotes we randomly selected 1,000 samples from the GrailQA and Simple Questions test set to constitute the testing set owing to the abundance of test samples.

| Model | Method | EM |
|---|---|---|
| Fine-Tuning | QGG (Query Graph Generator) (Lan & Jiang, 2020) | 44.1 |
|  | PullNet (Sun et al., 2019) | 45.9 |
|  | NSM+h (He et al., 2021) | 53.9 |
|  | CBR-KBQA (Das et al., 2021) | 67.1 |
|  | DecAF (Yu et al., 2023) | 70.4 |
| ChatGPT | KD-CoT (Wang et al., 2023b) | 49.2 |
|  | ToG | 57.1 |
|  | ToG-R | **58.9** |
| Llama2-70B-Chat | ToG | 53.6 |
|  | ToG-R | **57.6** |
| GPT-4 | ToG | 67.6 |
|  | ToG-R | **69.5** |

Table 10: The statics of Fine-Tuning, prompting-based methods of ComplexWebQuestions dataset.

| Model | Method | EM |
|---|---|---|
| Fine-Tuning | KD-CoT (Wang et al., 2023b) | 73.7 |
|  | NSM (He et al., 2021) | 74.3 |
|  | Program Transfer (Cao et al., 2022) | 74.6 |
|  | TIARA (Shu et al., 2022) | 75.2 |
|  | DecAF (Yu et al., 2023) | 82.1 |
| Code-davinci-002 | KB-BINDER (Li et al., 2023a) | 74.4 |
| ChatGPT | StructGPT (Jiang et al., 2023) | 72.6 |
|  | ToG-R | 75.8 |
|  | ToG | **76.2** |
| Llama2-70B-Chat | ToG-R | **69.4** |
|  | ToG | 64.1 |
| GPT-4 | ToG-R | 81.9 |
|  | ToG | **82.6** |

Table 11: The statics of Fine-Tuning, prompting-based methods of WebQSP dataset.

| Model | Method | EM |
|---|---|---|
| Fine-Tuning | DecAF (Yu et al., 2023) | 68.4 |
| | UniParser (Liu et al., 2022) | 69.5 |
| | TIARA (Shu et al., 2022) | 73.0 |
| | Pangu (Gu et al., 2023) | 75.4 |
| Code-davinci-002 | KB-BINDER (Li et al., 2023a) | 53.2 |
| ChatGPT | ToG-R | 66.4 |
| | ToG | **68.7** |
| GPT-4 | ToG-R | 80.3 |
| | ToG | **81.4** |

Table 12: The statics of Fine-Tuning, prompting-based methods of GrailQA dataset

| Model | Method | Acc |
|---|---|---|
| Fine-Tuning | SPARQL-QA(Santana et al., 2022) | 45.4 |
| ChatGPT | ToG-R | 48.6 |
| | ToG | **50.2** |
| GPT-4 | ToG | 53.8 |
| | ToG-R | **54.7** |

Table 13: The statics of Fine-Tuning, prompting-based methods of QALD10-en dataset.

| Model | Method | EM |
|---|---|---|
| Fine-Tuning | T5-LARGE+KPs (dos Santos et al., 2022) | 58.3 |
| | Memory Networks (Bordes et al., 2015) | 63.9 |
| | GETT-QA (Banerjee et al., 2023) | 76.1 |
| | DiFaR(Baek et al., 2023a) | **85.8** |
| ChatGPT | ToG-R | 45.4 |
| | ToG | **53.6** |
| GPT-4 | ToG-R | 58.6 |
| | ToG | **66.7** |

Table 14: The statics of Fine-Tuning, prompting-based methods of SimpleQuetsions dataset.

| Model | Method | EM |
|---|---|---|
| Fine-Tuning | T5.1.1-XXL+SSM (Raffel et al., 2020) | 43.5 |
| | PaLM (Chowdhery et al., 2022) | 43.5 |
| | RAG (Lewis et al., 2021) | 45.2 |
| | FiDO (de Jong et al., 2022) | 51.1 |
| | FiE+PAQ (Kedia et al., 2022) | 56.3 |
| PALM2 | Few-shot (Li et al., 2023a) | 28.2 |
| ChatGPT | BeamSearchQA$_{\text{Fine-tuned Retriever}}$ (Sun et al., 2023a) | 27.3 |
| | ToG-R | 53.2 |
| | ToG | **54.5** |
| GPT-4 | ToG-R | 57.1 |
| | ToG | **57.9** |

Table 15: The statics of Fine-Tuning, prompting-based methods of WebQuestions dataset.

| Model | Method | EM |
|---|---|---|
| Fine-Tuning | MetaRAG | 78.7 |
| | Wikipedia | 81.3 |
| | single ngram | 83.7 |
| | KGI_1 | 84.4 |
| | Re2G (Glass et al., 2022) | 87.7 |
| ChatGPT | ToG-R | 75.3 |
| | ToG | **76.8** |
| GPT-4 | ToG-R | 75.5 |
| | ToG | **77.1** |

Table 16: The statics of Fine-Tuning, prompting-based methods of T-REx dataset, where data are from the leaderboard.

| Model | Method | EM |
|---|---|---|
| Fine-Tuning | Multitask DPR + BART | 58.0 |
| | MetaRAG | 71.6 |
| | KGI_1 | 72.6 |
| | Wikipedia | 74.0 |
| | single ngram | 74.6 |
| ChatGPT | ToG-R | 86.5 |
| | ToG | **88.0** |
| GPT-4 | ToG-R | 86.9 |
| | ToG | **88.3** |

Table 17: The statics of Fine-Tuning, prompting-based methods of Zero-Shot RE, where data are from the leaderboard.

| Model | Method | EM |
|---|---|---|
| Fine-Tuning | RoBERTa-Large (Liu et al., 2019) | 80.6 |
| | T5-3B (Raffel et al., 2020) | 85.6 |
| | RACo-Large (Yu et al., 2022) | 88.2 |
| ChatGPT | ToG-R | **93.8** |
| | ToG | 91.2 |
| GPT-4 | ToG-R | 95.4 |
| | ToG | **95.6** |

Table 18: The statics of Fine-Tuning, prompting-based methods of Creak dataset.

## D CASE STUDY

In this section, we present a case analysis of the CWQ dataset to evaluate the utility and limitations of the ToG. We compared ToG with IO, CoT and the New Bing search engine[4]. We have selected four examples for analysis, each with top-3 reasoning paths and normalized scores.

In the first example in Table 19, ToG initially identifies "Arthur Miller" and "Lucian", in the question and subsequently expands its reasoning path through the Exploration and Reasoning processes. After conducting two iterations of the search, ToG successfully arrived at the correct answer, as it links the two entities with the reasoning path, which represents the perfect route for locating solutions. Additionally, the presence of *UnName_Entity* in the intermediate steps of reasoning paths, reflects the incompleteness of the knowledge graph (i.e., some entities lack the "name" relation). However, ToG is still capable of performing the next reasoning step, as all available relations contain relevant information. We observe that IO and CoT do not answer the query correctly since they lack the appropriate knowledge, and New Bing do not retrieve the appropriate information during the retrieval process.

In the second example shown in Table 20, IO prompt and CoT even New Bing suffer from a hallucination issue and provide an erroneous answer, "Florida", since the "Renegade" is the mascot of "Florida State Seminoles" instead of "fight song". ToG obtain the reasoning path "Renegade" → "sports.fight_song.sports_team" → "Pittsburgh Steeler". However, this reasoning path does not lead to a final answer, but combined with LLMs', ToG can answer the correct answer "Pennsylvania".

The third example in Table 21 demonstrates an example of the ToG-R, where ToG ignores the intermediate entities and focuses on the information in the relations instead. After two-hop of reasoning to "Harvard College", combined with LLMs', ToG gives the final result: "Massachusetts". It can be observed that IO and CoT do not have background knowledge, and New Bing answers the question correctly since it retrieves the correct information.

The final example is shown in Table 22. Where ToG generates a reasoning path to the final question (Path 1). Notably, the Ground-Truth reasoning path for the answer is *sports.sports_team.team_mascot* → *base.schemastaging.team_training_ground_relationship.facility*

---

[4]Accessed version July 2023.

→ *base.schemastaging.sports_team_extra.training_ground* (retrievable from the SPARQL), which is more hop than ToG. The ToG enables the exploration of new reasoning paths to reach the correct answer, which represents a significant application of knowledge graph reasoning. However, the answer to the current question in the KB, is "Bright House Field", which is incorrect since "Philadelphia Phillies" training stadium is "Spectrum Field" now. This example exemplifies a constraint of ToG, specifically its dependence on the correctness of the KB, where the incorrect KB has negative impact on ToG's reasoning accuracy. However, as depicted in Figure 4, ToG presents a novel framework to construct automated knowledge infusion to the KG.

| Question | Who influenced Arthur Miller that was influenced by Lucian? |
|---|---|
| Reasoning Paths | Arthur Miller → *influence.influence_node.influenced_by* → William Shakespeare → *influence.influence_node.influenced_by* → Lucian. (**Path 1**, Score: 0.75) |
| | Lucian → *influence.influence_node.influenced_by* → Socrates → *influence.influence_node.influenced_by* → Parmenides. (**Path 2**, Score: 0.2) |
| | Arthur Miller → *people.person.education* → **UnName_Entity** → *education.education.student* → Arthur Miller. (**Path 3**, Score: 0.05) |
| Outputs | **IO**: Edmund Wilson. |
| | **CoT**: Arthur Miller was influenced by the playwright and director Thornton Wilder, who in turn was influenced by Lucian of Samosata. |
| | **New Bing**: I'm not sure about Lucian's influence on Arthur Miller. However, Arthur Miller was influenced by Henrik lbsen and Eugene O'Neill. He was also influenced by the events of the McCarthy era and the Salem witch trials. |
| | **ToG**: Based on the given knowledge triples, we can infer that Arthur Miller was influenced by William Shakespeare, who was directly influenced by Lucian. |
| Ground Truth | William Shakespeare. |

Table 19: Case for ToG. The wrong answers are highlighted with red color, and the correct answers are highlighted with blue color.

| Question | What is the state where the team whose fight song is "Renegade" is from? |
|---|---|
| Reasoning Paths | Renegade → *sports.fight_song.sports_team* → Pittsburgh Steelers. (**Path 1**, Score: 0.67) |
| | Renegade → *sports.sports_team.fight_song* → **UnName_Entity**. (**Path 2**, Score: 0.33) |
| Outputs | **IO**: Florida. |
| | **CoT**: The team whose fight song is "Renegade" is the Florida State Seminoles, and they are from the state of Florida in the United States. |
| | **New Bing**: The team whose fight song is "Renegade" is the Florida State Seminoles. They are from the state of Florida in the United States. |
| | **ToG**: Based on the given knowledge triples, the team with the fight song "Renegade" is the Pittsburgh Steelers. The Pittsburgh Steelers are a professional American football team, and they are from the state of Pennsylvania. |
| Ground Truth | Pittsburgh Pennsylvania. |

Table 20: Case for ToG. The wrong answers are highlighted with red color, and the correct answers are highlighted with blue color.

| Question | What state is the college that Sampson Salter Blowers is a grad student of located? |
|---|---|
| Reasoning Paths | Sampson Salter Blowers → *education.education.student* → *education.education.institution* → {Harvard College}. (**Path 1**, Score: 0.75) |
| | Sampson Salter Blowers → *education.education.student* → *education.educational_institution.students_graduates* → {}. (**Path 2**, Score: 0.2) |
| | Sampson Salter Blowers → *education.education.student* → *people.person.education* → {}. (**Path 3**, Score: 0.05) |
| Outputs | **IO**: Unknown. |
| | **CoT**: I'm sorry, but I do not have enough information to answer your question. Can you please provide more context or details about Sampson Salter Blowers and the college he attended? |
| | **New Bing**: I'm sorry but I couldn't find any information about Sampson Salter Blowers being a grad student. However, he was a noted North American lawyer, Lovalist and jurist from Nova Scotia who graduated from Harvard College with a ba in 1763 and an ma in 1765. He was admitted as an attorney in the Suffolk lnferior Court in 1766. |
| | **ToG**: Based on the given relations chains, Sampson Salter Blowers is a grad student of Harvard College. Therefore, the answer to the question is Harvard College, which is located in Massachusetts. |
| Ground Truth | Massachusetts. |

Table 21: Case for ToG-R. "Unknown" denotes LLM is unable to generate a valid response. The wrong answers are highlighted with red color, and the correct answers are highlighted with blue color.

| Question | What is mascot Phillie Phanatic's team's spring training stadium? |
|---|---|
| Reasoning Paths | Phillie Phanatic → *sports.sports_team.team_mascot* → Philadelphia Phillies → *sports.sports_team.arena_stadium* → Bright House Field. (**Path 1**, Score: 0.8) |
| | Phillie Phanatic → *sports.sports_team.team_mascot* → Philadelphia Phillies → *ports.sports_team_location.teams* → Philadelphia. (**Path 2**, Score: 0.15) |
| | Phillie Phanatic → *sports.sports_team.team_mascot* → Philadelphia Phillies → *sports.sports_team.team* → Phillie Phanatic. (**Path 3**, Score: 0.05) |
| Outputs | **IO**: Spectrum Field. |
| | **CoT**: First, the mascot Phillie Phanatic represents the Philadelphia Phillies baseball team. Second, the Philadelphia Phillies' spring training stadium is located in Clearwater, Florida and is known as Spectrum Field. The answer is Spectrum Field. |
| | **New Bing**: The Philadelphia Phillies' spring training stadium is called BayCare Ballpark. |
| | **ToG**: The spring training stadium for Phillie Phanatic's team, the Philadelphia Phillies, is Bright House Field. |
| Ground Truth | Bright House Field (KB), Spectrum Field (Actual). |

Table 22: Case for ToG. The wrong answers are highlighted with red color, and the correct answers are highlighted with blue color.

# E    SPARQL AND PROMPTS

In this section, we show all the prompts that need to be used in the main experiments. First, we pre-define SPARQL for Freebase queries, which can be executed by simply filling in the appropriate mid and relation. For Wikidata, we abstain from employing executable SPARQL, rather we directly engage in querying through nine pre-defined service APIs.

## E.1    PRE-DEFINED SPARQL

### E.1.1    RELATION SEARCH

```
PREFIX ns: <http://rdf.freebase.com/ns/>
SELECT ?relation
WHERE {
    ns:mid ?relation ?x .
}

PREFIX ns: <http://rdf.freebase.com/ns/>
SELECT ?relation
WHERE {
    ?x ?relation ns:mid .
}
```

### E.1.2    ENTITY SEARCH

```
PREFIX ns: <http://rdf.freebase.com/ns/>
SELECT ?tailEntity
WHERE {
    ns:mid ns:relation ?tailEntity .
}

PREFIX ns: <http://rdf.freebase.com/ns/>
SELECT ?tailEntity
WHERE {
    ?tailEntity ns:mid ns:relation  .
}
```

### E.1.3    CONVERT MID TO LABEL

```
PREFIX ns: <http://rdf.freebase.com/ns/>
SELECT DISTINCT ?tailEntity
WHERE {
{
    ?entity ns:type.object.name ?tailEntity .
    FILTER(?entity = ns:mid)
}
UNION
{
    ?entity <http://www.w3.org/2002/07/owlsameAs> ?tailEntity .
    FILTER(?entity = ns:mid)
}
}
```

## E.2    PRE-DEFINED APIS

```
def label2qid(self, label: str) -> str:

def label2pid(self, label: str) -> str:

def pid2label(self, pid: str) -> str:

def qid2label(self, qid: str) -> str:

def get_all_relations_of_an_entity(self, entity_qid: str)
    -> tp.Dict[str, tp.List]:

def get_tail_entities_given_head_and_relation(self, head_qid: str,
    relation_pid: str)
    -> tp.Dict[str, tp.List]:

def get_tail_values_given_head_and_relation(self, head_qid: str,
    relation_pid: str) -> tp.List[str]:

def get_external_id_given_head_and_relation(self, head_qid: str,
    relation_pid: str) -> tp.List[str]:

def mid2qid(self, mid: str) -> str:
```

### E.3    TOG

#### E.3.1    RELATION PRUNE

Please retrieve $k$ relations (separated by semicolon) that contribute to the question and rate their contribution on a scale from 0 to 1 (the sum of the scores of $k$ relations is 1).

```
In-Context Few-shot
```

Q: {Query}

Topic Entity: {Topic Entity}

Relations: {list of relations}

A:

#### E.3.2    ENTITY PRUNE

Please score the entities' contribution to the question on a scale from 0 to 1 (the sum of the scores of all entities is 1).

```
In-Context Few-shot
```

Q: {Query}

Relation: {Current Relation}

Entites: {list of entities}

Score:

#### E.3.3    REASONING

Given a question and the associated retrieved knowledge graph triples (entity, relation, entity), you are asked to answer whether it's sufficient for you to answer the question with these triples and your knowledge (Yes or No).

```
In-Context Few-shot
```

Q: {Query}

Knowledge triples: {Explored Paths}

A:

### E.3.4 GENERATE

Given a question and the associated retrieved knowledge graph triples (entity, relation, entity), you are asked to answer the question with these triples and your own knowledge.

`In-Context Few-shot`

Q: {Query}

Knowledge triples: {Explored Paths}

A:

## E.4 ToG-R

### E.4.1 REASONING

Please answer the question using Topic Entity, Relations Chains and their Candidate Entities that contribute to the question, you are asked to answer whether it's sufficient for you to answer the question with these triples and your knowledge (Yes or No).

`In-Context Few-shot`

Q: {Query}

Topic Entity, with relations chains, and their candidate entities: {Explored Relation Chains}

A:

## E.5 CoT AND IO

### E.5.1 CoT PROMPT

Q: What state is home to the university that is represented in sports by George Washington Colonials men's basketball?

A: First, the education institution has a sports team named George Washington Colonials men's basketball in is George Washington University , Second, George Washington University is in Washington D.C. The answer is Washington, D.C.

Q: Who lists Pramatha Chaudhuri as an influence and wrote Jana Gana Mana?

A: First, Bharoto Bhagyo Bidhata wrote Jana Gana Mana. Second, Bharoto Bhagyo Bidhata lists Pramatha Chaudhuri as an influence. The answer is Bharoto Bhagyo Bidhata.

Q: Who was the artist nominated for an award for You Drive Me Crazy?

A: First, the artist nominated for an award for You Drive Me Crazy is Britney Spears. The answer is Jason Allen Alexander.

Q: What person born in Siegen influenced the work of Vincent Van Gogh?

A: First, Peter Paul Rubens, Claude Monet and etc. influenced the work of Vincent Van Gogh. Second, Peter Paul Rubens born in Siegen. The answer is Peter Paul Rubens.

Q: What is the country close to Russia where Mikheil Saakashvii holds a government position?

A: First, China, Norway, Finland, Estonia and Georgia is close to Russia. Second, Mikheil Saakashvii holds a government position at Georgia. The answer is Georgia.

Q: What drug did the actor who portrayed the character Urethane Wheels Guy overdosed on?

A: First, Mitchell Lee Hedberg portrayed character Urethane Wheels Guy. Second, Mitchell Lee Hedberg overdose Heroin. The answer is Heroin.

Q: {Query}

A:

### E.5.2 IO PROMPT

Q: What state is home to the university that is represented in sports by George Washington Colonials men's basketball?

A: Washington, D.C.

Q: Who lists Pramatha Chaudhuri as an influence and wrote Jana Gana Mana?

A: Bharoto Bhagyo Bidhata.

Q: Who was the artist nominated for an award for You Drive Me Crazy?

A: Jason Allen Alexander.

Q: What person born in Siegen influenced the work of Vincent Van Gogh?

A: Peter Paul Rubens.

Q: What is the country close to Russia where Mikheil Saakashvii holds a government position?

A: Georgia.

Q: What drug did the actor who portrayed the character Urethane Wheels Guy overdosed on?

A: Heroin.

Q: {Query}

A:

