# OpenReview forum: "Think-on-Graph: Deep and Responsible Reasoning of Large Language Model on Knowledge Graph"
_ICLR.cc/2024/Conference — ICLR 2024 poster_

### Official Review · Reviewer_LtPr · 2023-10-30

**Soundness:** 3 good
**Presentation:** 4 excellent
**Contribution:** 4 excellent
**Rating:** 8
**Confidence:** 4

**Summary:**

This paper presents an LLM-KG integration paradigm to incorporate structural knowledge stored in KGs in LLMs reasoning, namely Think-on-Graph (ToG). ToG makes LLM serve as an agent to walk on KGs by iteratively searching and pruning relations and entities from KGs. Experiments show that ToG could enhance the LLM’s reasoning capabilities and achieve SOTA on 6 datasets. ToG also exhibits knowledge traceability and correctability to improve KG quality.

**Strengths:**

1. Good written paper, very easy to read.
2. Strong experimental performance, achieving SOTA on 6 datasets without training.
3. The motivation is strong and clear, incorporating external knowledge (KGs) would be an important problem to enhance LLMs.

**Weaknesses:**

1. I believe it would be interesting to see the ToG performance when encountering the knowledge conflict between external KG knowledge and parametric knowledge stored in LLMs, which is an aspect to test the robustness of ToG.
2. I am concerned that ToG would bring too many intermediate steps and cause high latency in reasoning and expensive deployment, especially when using API-based black-box models like GPT-4.
3. It would be interesting to see the performance of ToG when incorporating the KGs of low quality (sparsity, noisy, etc), since most KGs are sparse and out of date to some extent. So investigating the impact of KG quality would enhance the understanding of ToG method and its limitations.
4. I would encourage authors to present experiment results on small LLMs, such as 7B and 13B.

**Questions:**

N/A

---

> ### Author Response · Authors · 2023-11-20
> **Response to Reviewer LtPr**
>
> Thank you so much for your comments and inspiring suggestions. Our reply to your comments and questions are listed below.
>
> **1.	Reviewer's Comment**: I believe it would be interesting to see the ToG performance when encountering the knowledge conflict between external KG knowledge and parametric knowledge ...
>
> **Reply**: We thank the reviewer for your inspiring comment on what happens when KG conflicts with LLM. Although the intrinsic knowledge of LLMs is difficult to probe, we can use the reasoning results of LLM and the results of the KG queries to determine whether there is a knowledge conflict between LLM and KG instead. We run an experiment by analyzing the log file of ToG experiment on CWQ dataset and count KG's performance when LLM and KG output different answers. Among the questions that CoT with GPT-4's (representing LLM) answer conflicts with KG's answer, the accuracy of ToG with GPT-4 is 43.4%, a 13.7% reduction from its performance on CWQ 57.1%.
>
> **2.	Reviewer's Comment**: I am concerned that ToG would bring too many intermediate steps and cause high latency in reasoning and expensive deployment
>
> **Reply**: Please refer to our submitted official comment "Response to All reviewers about the concerns on the Efficiency of ToG", where we elaborate on the maximum cost of answering a question needed by our method and the average cost on CWQ. We also propose some feasible solutions on how to improve the efficiency of ToG (some have been empirically verified in experiments). In particular, we show in this official comment that the latency in inference is NOT as high as what people may imagined using improved ToG algorithms. In most cases, the number of LLM calling by ToG is only single digit (even <5 times in most cases).
>
> **3.	Reviewer's Comment**: It would be interesting to see the performance of ToG when incorporating the KGs of low quality (sparsity, noisy, etc.), since most KGs are sparse and out of date to some extent.
>
> **Reply**:
>
> (1)	We agree with the reviewer that quality of knowledge graphs may have significant impact on the performance of ToG. We are doing experiments to test how the qualify of a KG affect ToG's performance. Specifically, we artificially add different amount of noise/errors and/or delete different amount of triples existing in original wikidata, and evaluate how much ToG's performance drops down. Unfortunately, the experiments are still in progress due to limited computational and LLM token resource. We are trying our best to show the experimental results asap.
>
> (2)	Another important task we are doing is to research potential solutions for alleviating the negative effect of low-quality KG in ToG inference. Firstly, in Section 3.3, we have illustrated how ToG could correct the wrong information in KG with the help of LLMs and users (identifying potential errors in KG by LLM and determining the correctness by human). More concretely, the LLM can suggest to the users which knowledge in the reasoning paths may be wrong, and the users reply in the system on whether execute a correction action. Secondly, we are studying how to improve the quality of KG by fusing diverse external information source such as variant LLMs, search engine and many others.
>
> **4.	Reviewer's Comment**: I would encourage authors to present experiment results on small LLMs, such as 7B and 13B.
>
> **Reply**: Following the reviewer's suggestion, we tested the performance using LLaMA2-7b and LLaMA2-13b, two very small LLMs, on WebQSP, and the results are summarized below.
>
> | Method | LLaMA2-7b | LLaMA2-13b |
> | --- | --- | --- |
> | CoT | 7.1%| 8.6% |
> | ToG | 12.6% | 25.1% |
>
> Using exact match as the evaluation metric, the CoT prompting with LLaMA-7B and LLaMA-13B can only achieve 7.1% and 8.6%, respectively, on WebQSP. Meanwhile, ToG with LLaMA2-7B and LLaMA2-13B can achieve 12.6% and 25.1%, respectively. Consistent with our expectation, ToG performs better than CoT with these two very small LLMs. However, the actual accuracies for CoT and ToG are indeed far below from our expectation. After careful investigation, we found the main factor of the bad performance is: since both LLaMA2-7B and LLaMA2-13B are base models which have not been trained with supervised fine tuning and RLHF, the format of their outputs are really difficult to manage and difficult to exactly match the format required for a "correct" answer. For example, we require the answers from LLM should be put between { and }, but these two small LLMs always violate this rule and make the evaluator difficult to locate the answer, leading its answer misclassified to "wrong". In another example, we ask LLM to prune and return top 3 relations from candidates (A, B, C, D, E), but LLaMA2-7B and LLaMA2-13B return irrelevant relations, e.g., W, S, T etc.  In some cases, LLaMA2 models restate the input questions or repeat the text over and over again without responding to the question.

---

> ### Author Response · Authors · 2023-11-23
> **Kind Request for Timely Feedback on our Responses**
>
> Dear Reviewer,
>
> We are writing to follow up on the response we submitted regarding your comments.
>
> Firstly, we would like to express my gratitude for your initial review and valuable feedback. The insights provided have been instrumental in refining our paper, and we have tried our best to address each point in our response with the aim of enhancing the paper's quality and relevance.
>
> Understanding the time and effort required for a thorough review, we greatly appreciate the commitment you have shown towards ensuring the high standards of ICLR. However, as the rebuttal process is concluding, we are keenly awaiting your further feedback on our responses. The comments from your expertise are crucial for the final preparation and improvement of our paper.
>
> If you have any further questions or require more information to raise your initial score, please feel free to let us know. We are fully committed to making all necessary adjustments to meet your expectation.
>
> Thank you once again for your time and dedication. We look forward to your valuable feedback.

---

### Official Review · Reviewer_K4jf · 2023-10-31

**Soundness:** 3 good
**Presentation:** 3 good
**Contribution:** 3 good
**Rating:** 8
**Confidence:** 4

**Summary:**

This paper presents a novel approach called Think-on-Graph (ToG) to synergize the LLMs and KGs for reasoning. ToG enables LLMs as agents to iteratively execute searches on KGs to discover promising reasoning paths, which are then used to guide the LLMs to generate accurate answers. ToG is a general framework that can be applied to various LLMs and KGs. Extensive experiments on several datasets show that ToG can significantly improve the reasoning performance of LLMs.

**Strengths:**

1. This paper is well-presented and easy to follow. The authors provide a clear motivation and a good introduction to the problem.

2. The proposed framework can be easily plugged into existing LLMs and KGs without incurring additional training costs.

3. Extensive experiments on several datasets show that ToG can significantly improve the reasoning performance of LLMs.

**Weaknesses:**

1. The computational cost of ToG is relatively high. The searching process of ToG involves multiple LLM calls, which may be costly and limit its practical applicability in some settings.

2. Some details are inconsistent in the paper. For example, in the approach introduction section, ToG selects the next step triples/relations based on current expended reasoning paths. However, in the prompt illustrated in G.3, I cannot find the current reasoning paths used for the pruning process.

3. I have concerns that ToG might not understand the meaning of relations well and generalize to different KGs. The relations defined in KGs are usually in diverse formats. For example, the relations in Freebase are defined in a hierarchical format, while the relations in Yago have clearer semantics. If the relations do not reveal the clear semantics, the searching process of ToG might be misled.

**Questions:**

1. Can the authors discuss the cost of ToG in detail? What is the average number of calls for the reasoning? What is the overall price for the ChatGPT/GPT-4 API calls?

2. Can the authors explain the inconsistency I discussed above and present a clear illustration of the whole reasoning process?

3. Can the authors explain how ToG can generalize to different KGs? How LLMs in ToG understand the meaning of relations in different KGs without additional training?

4. Can other search algorithms be used for ToG? For example, depth-first search, breadth-first search, A* search, etc.

5. Can ToG be used to solve complex reasoning problems that cannot be solved by path-based reasoning? For example, in GrailQA, there are questions like: "How many TV programs has Bob Boyett created?". This question needs to count the KG structures to get the answer. Besides, there are other complex questions requiring intersection, union, and negation operations. Can ToG be used to solve these kinds of questions? Maybe a limitations section can be added to the paper.

---

> ### Author Response · Authors · 2023-11-20
> **Response to Reviewer K4jf**
>
> Thank you so much for your comments and inspiring suggestions. Our reply to your comments and questions are listed below.
>
> **1.Reviewer's Comment**: The computational cost of ToG is relatively high. The searching process of ToG involves multiple LLM calls, which may be costly and limit its practical applicability in some settings. Can the authors discuss the cost of ToG in detail?
>
> **Reply**:  Please refer to our submitted official comment "Response to All reviewers about the concerns on the Efficiency of ToG", where we elaborate on the maximum cost of answering a question needed by our method and the average cost on CWQ. We also propose some feasible solutions on how to improve the efficiency of ToG (some have been empirically verified in experiments).
>
> **2.Reviewer's Comment**: Some details are inconsistent in the paper. For example, in the approach introduction section, ToG selects the next step triples/relations based on current expended reasoning paths. However, in the prompt illustrated in G.3, I cannot find the current reasoning paths used for the pruning process.
>
> **Reply**: First of all, we are sorry for the confusion about what (whole path or only triple?) ToG really gives LLM's prompt in pruning. We clarify it here: ToG selects the next relations/entities based only on the literal information of candidate entities/relations and the given question (as mentioned in Section 2.1.2), rather than the current whole expanded reasoning paths. We would revise the description in the manuscript to make it clearer for readers. In addition, in future work, we are going to study whether the pruning process might be improved by inputting current expanded reasoning paths instead.
>
> **3.Reviewer's Comment**: I have concerns that ToG might not understand the meaning of relations well and generalize to different KGs. The relations defined in KGs are usually in diverse formats. For example, the relations in Freebase are defined in a hierarchical format, while the relations in Yago have clearer semantics.
>
> **Reply**: We understand the reviewer's concern whether LLM in ToG can correctly understand complex and diverse relations in various KGs. Conceptually, although the format of relations in Freebase is different from natural language, the relation names still consist of words and phrases understandable by LLMs (e.g., people.person.occupation, sports.mascot.team, the last word shows the meaning of relation). In the case of Wikidata, most relations are quite similar to natural language (e.g., instance of, part of, founded by). Empirically, it can be seen from our experimental results that LLMs and light-weight pruning models can perform relation pruning well on both Freebase and Wikidata. In addition, the reviewer's concern leads us to study whether ToG can be improved either by cleaning the relation names in KGs before inputting them to LLMs or by incorporating the specific relation structure of a KG. We would study this problem in the future.
>
> **4.Reviewer's Comment**: Can the authors discuss the cost of ToG in detail? What is the average number of calls for the reasoning? What is the overall price for the ChatGPT/GPT-4 API calls?
>
> **Reply**: Please refer to our submitted official comment "Response to All reviewers about the concerns on the Efficiency of ToG", where we elaborate on the maximum cost of answering a question needed by our method and the average cost on CWQ. We also propose some feasible solutions on how to improve the efficiency of ToG (some have been empirically verified in experiments).
>
> **5.Reviewer's Comment**: Can the authors explain the inconsistency I discussed above and present a clear illustration of the whole reasoning process?
>
> **Reply**: Since this question is similar to Comment 2, please refer to the reply to Comment 2.
>
> **6.Reviewer's Comment**: Can the authors explain how ToG can generalize to different KGs? How LLMs in ToG understand the meaning of relations in different KGs without additional training?
>
> **Reply**: This question is similar to Comment 3. Please refer to the reply to reviewer's comment 3.
>
> **7.Reviewer's Comment**: Can other search algorithms be used for ToG? For example, depth-first search, breadth-first search, A* search, etc.
>
> **Reply**: Yes, theoretically, the search algorithm in ToG can be replaced to many other search algorithms, depending on the trade-off between efficiency and sufficiency. For example, beam search can be considered as a constrained breadth-first search (BFS), in order to improve search efficiency by appropriately sacrificing sufficiency. Likewise, constrained DFS and A* algorithm can also be used for searching optimal reasoning paths of ToG. In the future work, we are studying which search algorithms are better and how to automatically select the best search algorithm for ToG according to specific task and specific KG structure. In addition, we are going to study new search algorithms to optimize the performance and efficiency of ToG.

---

> > ### Author Response · Authors · 2023-11-20
> > **Response to Reviewer K4jf**
> >
> > **8.Reviewer's Comment**: Can ToG be used to solve complex reasoning problems that cannot be solved by path-based reasoning? For example, in GrailQA, there are questions like: "How many TV programs has Bob Boyett created?". This question needs to count the KG structures to get the answer. Besides, there are other complex questions requiring intersection, union, and negation operations. Can ToG be used to solve these kinds of questions? Maybe a limitations section can be added to the paper.
> >
> > **Reply**: Although vanilla ToG may not solve these complex reasoning problems as mentioned by the reviewer, a slight variant of ToG can solve all these complex problems mentioned above. Specifically, if we swap the order of pruning and reasoning in each iteration of vanilla ToG, the new ToG algorithm (call new-ToG) can solve all above complex reasoning problems. For example, when new-ToG answers the question "How many TV programs has Bob Boyett created?", new-ToG searches all neighbors of "Bob Boyett" related to "TV programs" and generates $M$ reasoning paths:  {Bob Boyett-producer-program1, Bob Boyett-producer-program2, Bob Boyett-producer-program3... Bob Boyett-producer-programM}. LLMs can "count" the number of neighbors during the reasoning step and output the answer $M$. Note that vanilla ToG can't finish this task because these neighbors are pruned in each iteration and only top $N$ (width of beam search) are kept in next iteration, so it can't count all neighbors.
> > For intersection questions, such as "How many TV programs have Bob Boyett and somebody created together?" According to Bob Boyett - productor - {program1, program2, program3... programN} and someone - productor - {program1, program2, program3... programM} , LLM reasoner can output the count of intersections. Similarly, union and negation operations can be implemented by LLM reasoner if a complete neighbor list (before pruning) can be provided.

---

> > ### Comment · Reviewer_K4jf · 2023-11-20
> >
> > I thank the authors for providing additional experimental results and responses to my questions. The efficiency is a concern raised by many reviewers (myself included), and I welcome the discussions on how it can be improved, albeit with some sacrifice of task performance.
> >
> > The responses have addressed most of my questions, and I believe my original score is still a fair assessment of the paper. Therefore I will maintain my score.

---

> ### Author Response · Authors · 2023-11-23
> **Reply to Reviewer K4jf**
>
> We strongly recognize the need for a discussion on how to improve the efficiency of algorithms. In this paper, we have discussed some solutions such as the proposed ToG-R and the use of lightweight models instead of large language models as pruning models. We will also explore more ways to improve algorithmic efficiency afterwards, including some of the methods we mentioned in our submitted official comment **Response to all reviewers about the concerns on the efficiency of ToG**. We are also open to suggestions and comments for the efficiency issue.
>
> We are deeply appreciative of your  recognition of the paper's contribution. And your expertise and thoughtful review have made a significant impact on our research. We look forward to potentially incorporating more of your valuable insights in our future work.
>
> Thank you once again for your time and guidance.

---

### Official Review · Reviewer_ezze · 2023-10-31

**Soundness:** 3 good
**Presentation:** 2 fair
**Contribution:** 3 good
**Rating:** 6
**Confidence:** 4

**Summary:**

The paper proposes ToG (Think-on-Graph) for deep, responsible, and efficient LLM reasoning with knowledge graphs with a new paradigm of "LLM×KG." The ToG is with the searching and pruning procedures to conduct deep reasoning. Empirical results on five KBQA datasets with extensive experiments justify the effectiveness of the proposed ToG method.

**Strengths:**

The paper is rather solid and detailed from a technical perspective.

The figures are clear and easy to understand.

The solution of LLM×KG is reasonable and new to the LLM community.

The depth and width of Toc are clearly investigated in the ablation study.

The limitations of the proposed method are extensively discussed in Appendix A.

Several technical details, case studies, and evaluation results are also elaborated on in the Appendix.

**Weaknesses:**

The writing of the paper can be largely improved.

Besides, the mathematical notations and equations can be improved to be clearer.

It would be better to summarize the frequently used notations in one table or sentence.

The empirical performance of ToG is not consistently the best, which is outperformed by the "prior finetune SOTA" in some cases of Tab. 1. Although not requiring training is a major benefit of ToG, its reasoning power is not fully convincing enough. I would suggest the paper make a further discussion and explanation for that.

Besides, the important baseline, "Prior Prompting SOTA" in Tab.1, is not sufficiently evaluated and compared. It would be much better and more convincing to fill in the blanks in Tab.1.

The paper mentions the hallucination problem many times and uses it as the motivation of ToG. However, the hallucination problem is not sufficiently studied. As far as I can see, there is only one preliminary analysis of error is provided in Appendix D.2.

The paper is empirically driven and lacks in-depth analysis, whether from methodological or theoretical perspectives.

**Questions:**

What is the running-time efficiency of ToG?

How do the KGs used in the experiment part solve the limitation of out-of-date knowledge?

The beam search with pruning adopted by ToG is quite relevant to the progressive reasoning methods equipped with learnable search and pruning mechanisms, which also come from the KG areas (e.g., AdaProp [1] and Astarnet [2]). I would suggest the paper have a discussion with these relevant works. In addition, is it possible to achieve a kind of learnable pruning as an enhancement of ToG?

[1] Zhang et al. AdaProp: Learning Adaptive Propagation for Graph Neural Network based Knowledge Graph Reasoning. KDD 2023.

[2] Zhu et al. A*Net: A Scalable Path-based Reasoning Approach for Knowledge Graphs. NeurIPS 2023.

---

> ### Author Response · Authors · 2023-11-20
> **Response to Reviewer ezze**
>
> We would like to appreciate your valuable time and comments. Hope our following response can answer your questions.
>
> **1.	Reviewer's Comment**: "The writing of the paper can be largely improved. Besides, the mathematical notations and equations can be improved to be clearer. It would be better to summarize the frequently used notations in one table or sentence."
>
> **Reply**: We sincerely thank the reviewer's suggestions on paper writing. Firstly, we are improving the manuscript writing to make it easier to read. Secondly, we follow your suggestions to add a table explaining all notations in this paper and to refine all notations and equations to be clearer. All these revisions will be added to the final version of the manuscript.
>
> **2.	Reviewer's Comment**: "The empirical performance of ToG is not consistently the best, which is outperformed by the "prior finetune SOTA" in some cases of Tab. 1. Although not requiring training is a major benefit of ToG, its reasoning power is not fully convincing enough. I would suggest the paper make a further discussion and explanation for that."
>
> **Reply**: ToG doesn't outperform "prior finetune SOTA" on SimpleQuestion and T-REx, which both are single-hop reasoning datasets. As we claimed in the end of the first paragraph of Sec 3.2.1, one of ToG's advantage is its deep reasoning ability and thus it is narually good at multi-hop reasoning tasks rather than single-hop reasoning. Specifically, answering single-hop questions does not really need long reasoning path like what ToG does with LLM on KG. It is noticeable that, although Zero-Shot RE is also a single-hop dataset, ToG (88.3%) significantly outperforms the prior FT SOTA method (74.6%), which means ToG has chance to perform good on single-hop questions. In previous version of manuscript, ToG-R (69.5%) performs slightly worse than prior FT SOTA (70.4%) on multi-hop dataset CWQ, partially because we simply set search width $N=3$ and search depth $D=3$ without any comprehensive hyper-parameter tuning. During the rebuttal period, we rerun the experiment with larger $N=4$ and $D=4$, and the accuracy of ToG-R increases from the original 69.5% to the new 72.5%, exceeding the prior FT SOTA's 70.4%. We are going to update the data in the next version of this manuscript.
>
> **3.	Reviewer's Comment**: "Besides, the important baseline, "Prior Prompting SOTA" in Tab.1, is not sufficiently evaluated and compared. It would be much better and more convincing to fill in the blanks in Tab.1."
>
> **Reply**：We have the same concern with the reviewer. Although we have tried our best to search and collect "Prior Prompting SOTA" results from all investigated publications (incorporate external KGs into LLM reasoning), only few of them have been evaluated publicly on the datasets we used, and some of these prior works have not released their source codes publicly. That's why we only show "Prior Prompting SOTA" results on two datasets in Figure 1 when we submitted this manuscript. To implement a more comprehensive and sufficient comparison as you suggested, we are now reproducing the source codes of many other prompting-based methods and evaluating them on all datasets we used in this paper. The relevant experiments are still in progress, and we are sorry that new results can't be reported here due to limited time for rebuttal. We expect to be able to complete the evaluation experiments as soon as possible and update the Table 1 with the evaluation results in the later version of our paper.
>
> **4.	Reviewer's Comment**: "The paper mentions the hallucination problem many times and uses it as the motivation of ToG. However, the hallucination problem is not sufficiently studied. As far as I can see, there is only one preliminary analysis of error is provided in Appendix D.2."
>
> **Reply**: In this article, we don't aim to figure out the whole hallucination problem. Rather, we aim to solve a subset of the hallucination problem: accurate deep-reasoning problem, which account for a large amount of hallucination in LLM reasoning. From the results in Table 1, Table 2 and tables in appendix, we can see ToG brings significant accuracy improvement in deep-reasoning problems compared with LLM-only methods （CoT etc.).
>
> **5.	Reviewer's Comment**: What is the running-time efficiency of ToG?
>
> **Reply**: Please refer to our submitted official comment "Response to All reviewers about the concerns on the Efficiency of ToG".

---

> ### Author Response · Authors · 2023-11-20
> **Response to Reviewer ezze, specifically about the methodological and theoretical analysis**
>
> **6.	Reviewer's Comment**: "The paper is empirically driven and lacks in-depth analysis, whether from methodological or theoretical perspectives."
>
> **Reply**: Along with the algorithmic research work of ToG, we are studying a theoretical framework for ToG, and even more general, for knowledge-driven LLM. We will add preliminary results of theoretical analysis in the camera-ready version of this manuscript (appendix) if this paper is accepted and a more comprehensive theoretical framework and mathematical analysis will be published in future articles.
>
> We briefly highlight the main results of theoretical analysis as follows. A more detailed version of theoretical description can be found at this anonymous document: https://imgbox.com/hI5D5FSu
>
> (1) We reformulate the knowledge-driven LLM problem (where ToG is a special case) as a two-level nested Markov decision process (and thus can be analyzed via RL theory), where the outer loop is the interaction between LLM and human user and the inner loop is the interaction between LLM and external knowledge base such as KG. An illustrative figure can be found at https://imgbox.com/hYXybh91.
>
> (2) In the inner loop, the reasoner LLM first checks the contents in memory buffer (which we denote as information state) and generate a reasoning action consisting of both knowledge base (KB, instantiated as KG in ToG) query (correspond to the `search` step in ToG) and information filtering (correspond to the `pruning` step in ToG). The KB, serving as the environment, takes the query action and updates the memory buffer content (correspond to transition function in Markov decision process (MDP)). The memory buffer content is also updated deterministically via `pruning`. A judge (instantiated as LLM in ToG) then takes memory buffer content and evaluate its informativeness in terms of answering user question (correspond to the `reasoning` step in ToG), serving as the reward function in MDP. The (information, action, reward) tuple is updated to the memory buffer, and the accumulated memory buffer content serves as information state for the next iteration.
>
> (3) The goal of this MDP (RL problem) is to find a reasoning policy that maximizes its value function: $$V^\pi_\theta(s) = \mathbb{E}\bigl[\sum_{t=0}^\infty \gamma^t r_\theta(s_t, a_t) | s_0=s\bigr]$$
> where $\pi$ is the reasoning policy induced by LLM inference. Notably, LLM inference can be considered as performing implicit Bayesian inference [1], where LLM infers the hidden 'concept' contained in the prompts a posteriori, and then generates output based on the concept and prompts. Formally, this process can be written as:
> $$ p(y|x) = \int_\theta p(y|\theta, x) p(\theta|x) d\theta,$$ where $\theta$ parametrizes the knowledge environment (i.e. $\theta$ is the underlying knowledge).
> For our task of LLM reasoning, the `concept` corresponds to the underlying knowledge that can be inferred from knowledge prompts. In this sense, the induced policy $\pi$ can be considered as Bayesian model-based planning (ToG belongs to this type as a special case), where the LLM reasoner first implicitly estimates the posterior distribution of knowledge ($\theta$) relevant to user problem (based on prior obtained via pretraining and data obtained via KB interaction), and then uses $\theta$ to implicitly parameterize its internal world model and performs model based planning. In this sense, the `pruning` step in ToG can be regarded as a tree search planning algorithm with 0 lookahead step.
>
> (4) Assuming no bias between the LLM prior ($p(\theta)$) and the knowledge prior ($p(\theta^*)$), then the gap between an LLM-induced reasoning policy $\pi$ and an optimal reasoning policy, results only from the uncertainty in posterior estimation of the model parameter, $p(\theta|s_t)$. Since the uncertainty gradually decreases as information accumulates, we can use Bayesian regret [3], R(T), to measure the asymptotic performance of the reasoning policy $\pi$:
> $ \mathcal{R}(T) = \mathbb{E}\_{\theta \sim p(\theta)} \bigl[\sum_{t=1}^T V^{\pi^*}\_\theta (s_t) - V^{\pi^t}\_\theta (s_t) \bigr]$
> Utilizing proofs in [2] we can guarantee that this regret for LLM x KG is sublinear to the number of reasoning steps $T$ (so $\lim_{T \rightarrow \infty} \frac{R(T)}{T} = 0$), indicating that the reasoning will eventually converge to the correct information/answer due to the continuous uncertainty reduction in posterior inference $p(\theta|s_t)$ thanks to the iterative information gain from KB interaction.
>
> (5) The analysis of the outer user feedback loop can be conducted in a similar way, and the analysis suggests that the response precision can also improve with iterative user feedback, as demonstrated in Figure. 4 of this submission.

---

> > ### Author Response · Authors · 2023-11-20
> > **Response to Reviewer ezze, specifically about the theoretical analysis, out-of-date knowledge in KGs and learnable search algorithms**
> >
> > (6) The effect of each individual components in this framework, such as KB types, reasoner types (LLM vs. rules), can also be discussed in the context of Bayesian inference. Specifically, a structured KB (such as KG used in ToG) provides less noisy environment observations, favoring the uncertainty reduction in posterior inference. And utilizing LLM as the reasoner is critical to convergence, since otherwise no Bayesian inference mechanism can be formulated. Besides, compared with LLM x KG, the LLM + KG framework has no theoretical performance guarantee (in terms of Bayesian regret), since it is one-shot and cannot converge to accurate posterior estimation without iterative knowledge interactions.
> >
> > For detailed analysis, please refer to the document mentioned above.
> >
> > References:
> >
> > [1]. Xie et al., An Explanation of In-context Learning as Implicit Bayesian Inference, https://openreview.net/forum?
> > id=RdJVFCHjUMI
> >
> > [2]. Liu et al., Reason for Future, Act for Now: A Principled Framework for Autonomous LLM Agents with Provable Sample Efficiency, http://arxiv.org/abs/2309.17382
> >
> > [3]. Tor Lattimore, Csaba Szepesvari, Bandit Algorithms, Cambridge University Press, July 2020
> >
> > **7.	Reviewer's Comment**: How do the KGs used in the experiment part solve the limitation of out-of-date knowledge?
> >
> > **Reply**:  Firstly, the knowledge in KG is much easier to be updated than the knowledge in LLM. Specifically, the knowledge in KGs is stored on entities and relations (in the form of triples) which are usually physically stored on graph databases, and can be added, deleted and updated immediately. On the contrary, updating knowledge in LLM needs training with new samples which is usually time-consuming. Secondly, in practical scenarios, many KGs are updated frequently or even in real-time by adding new knowledge and deleting or revising out-of-date knowledge. The knowledge updating process can be automatically performed by a knowledge extraction system or manually done with crowdsourcing. For example, the knowledge stored in Wikidata is extracted automatically from Wikipedia and updated regularly according to the changes in Wikipedia content edited by human. Therefore, the knowledge stored in KGs such as Wikidata often has better timeliness than the inner knowledge of LLMs. Thirdly, the version of Wikidata used in this manuscript was released in Jan. 2023 and the latest version was released in Nov. 2023 where the newest knowledge is updated to Wikipedia. In comparison, the cut-off date of the knowledge of ChatGPT is September 2021, about two years ago.
> >
> > **8.	Reviewer's Comment**: The beam search with pruning adopted by ToG is quite relevant to the progressive reasoning methods equipped with learnable search and pruning mechanisms, which also come from the KG areas (e.g., AdaProp [1] and Astarnet [2]). I would suggest the paper have a discussion with these relevant works. In addition, is it possible to achieve a kind of learnable pruning as an enhancement of ToG?
> >
> > **Reply**:  We sincerely thank the reviewer's good suggestion about learnable search and pruning. Firstly, following your suggestion, we will add discussion relevant works such as AdaProp and Astarnet. Secondly, we are studying adding learnable pruning algorithms to improve ToG, and hopefully well-studied new algorithms can be used in next version of ToG. Again, we appreciate your inspiring suggestion so much.

---

> ### Comment · Reviewer_ezze · 2023-11-23
>
> Thanks for providing the responses that have addressed most of my concerns. I would suggest the authors add the extra analysis and discussion to the draft and improve the presentation as promised. I will retain my score and suggest an acceptance.

---

> ### Author Response · Authors · 2023-11-23
> **Response to Reviewer ezze**
>
> We would like to express my sincere gratitude for your valuable feedback on this work and for agreeing to its acceptance.
>
> Your insights have been instrumental in enhancing the quality of my work. We have diligently addressed the concerns you raised in your initial review, specifically about the complexity analysis and theoreotical analysis of ToG. We believe these revisions will significantly improve the paper, aligning it more closely with the conference's standards and expectations.
>
> In light of these substantial changes, we kindly request you to reconsider the score of 6 out of 10. We feel that the improvements made in response to your feedback have elevated the paper's quality and contribution. If there are specific areas in the paper that you feel still need improvement or if our responses have not fully addressed your initial concerns, we would greatly appreciate your further guidance. We are committed to achieving the highest standard for our work and are open to any additional suggestions you might have.
>
> Thank you once again for your time and consideration. Your expertise and guidance are greatly appreciated.

---

### Official Review · Reviewer_nifq · 2023-11-11

**Soundness:** 3 good
**Presentation:** 3 good
**Contribution:** 3 good
**Rating:** 6
**Confidence:** 4

**Summary:**

This work aims at improving the integration of knowledge graphs (KGs) in LLMs. The authors propose a method called Think-on-Graph (ToG) that performs beam-search on knowledge graphs, keeping track and exploring reasoning paths. Specifically, they use a "search" operation backed by the KG and a "prune" operation backed by the LLM. Through the experiments on benchmarks of KBQA/open QA/etc, the authors find an advantage of ToG in several of them compared to prior work. They also perform analyses on the effect of individual components of ToG like the selection of KG, search depth, pruning method, etc.

**Strengths:**

This paper proposes an intuitive and novel method in LLM-KG integration. The experiments are performed on a variety of datasets. The performance of the method is overall positive. The analyses on each component of the method is extensive. The writing of the paper is overall clear.

**Weaknesses:**

The reliance on very strong (production-level) LLMs and the choice of baselines. The authors explored the use of Llama-2-chat (70b), ChatGPT, and GPT-4 as the LLM in the ToG method. From Table 1 and Table 2, it seems that the strength of the LLM is essential to the method (Llama worse than ChatGPT, ChatGPT much worse than GPT-4). And the performance advantage on 6 out of 9 benchmarks is only observed with GPT-4. Additionally, though the authors show GPT-4 benefits from ToG compared to non-KG prompting (e.g., CoT), stronger prompting methods targeting for compositionality may be investigated, for example, self-ask [1]. Web search and vanilla retrieval augmentation methods can also be investigated [2].

Efficiency of the method. The process of beam search and pruning with LLMs can be costly. An extensive comparison on decoding time and cost across all methods should be performed and discussed.

[1] Press et al. 2022. Measuring and Narrowing the Compositionality Gap in Language Models. https://arxiv.org/abs/2210.03350
[2] Kasai et al. 2022. RealTime QA: What's the Answer Right Now? https://arxiv.org/abs/2207.13332

**Questions:**

Please see the weaknesses section.

---

> ### Author Response · Authors · 2023-11-19
> **Response to Reviewer nifq**
>
> Thank you for your insightful comments and suggestions. We appreciate the opportunity to address your concerns and questions.
>
> **1.Reviewer's Comment**: "The reliance on very strong (production-level) LLMs and the choice of baselines. The authors explored the use of Llama-2-chat (70b), ChatGPT, and GPT-4 as the LLM in the ToG method. From Table 1 and Table 2, it seems that the strength of the LLM is essential to the method (Llama worse than ChatGPT, ChatGPT much worse than GPT-4). And the performance advantage on 6 out of 9 benchmarks is only observed with GPT-4. ".
>
> **Reply**: We agree with the reviewer that the performance of ToG relies on the foundation LLM since the LLM plays an important role in the pruning and reasoning steps of ToG. However, we would emphasize the particular advantages of ToG:
>
> (1) ToG with small and poor foundation LLM has an opportunity to perform better than large and powerful LLM-only model, as shown in Table 2 of the manuscript. We can see that, on dataset CWQ, ToG-R with LLama2-70B-Chat reaches an accuracy 57.6%, higher than the accuracy of GPT-4 (Chain-of-thought prompting) 46.0%. Similarly, on dataset WebQSP, ToG-R with LLama2-70B-Chat reaches 68.9%, higher than GPT-4 (Chain-of-thought prompting) 67.3%. This result exhibits that ToG has a significant advantage in low cost deployment. Specifically, ToG with a cheaper and faster small LLM may be good enough in many scenarios, rather than deploying expensive and slow large LLM.
>
> (2) Although ToG relies on the foundation LLM, its performance also benefits from the contribution of knowledge graph, as well as the interaction of LLM and KG. We can see from Table 2 that, for any specific LLM, ToG always outperforms the corresponding LLM-only model. E.g., on dataset CWQ, ToG with ChatGPT reaches 58.9%, and ChatGPT-only (CoT) reaches 38.8%, so there is a 20.1% accuracy gap. We would claim that, ToG has the flexibility of plug-and-play any LLMs and KGs, and this algorithm usually improve the performance of the original LLM by incorporating the structural knowledge from KGs without adding any additional training cost (training-free).
>
> (3) Please allow us to emphasize that, all prior SOTA methods belong to training-based methods, which has a natural advantage in prediction accuracy on specific task or specific dataset compared with prompting-based methods (ToG, CoT etc.). However, ToG, a prompting-based method, outperforms those training-based prior SOTA methods on 6 out of 9 datasets. Both LLM and KG contribute to the performance of ToG (from Table 2, we see GPT-4 only is not as good as prior SOTAs without the help of KG). Moreover, we would also emphasize that, prompting-based methods have their own particular advantages, such as avoid of additional training cost and better explainability.
>
> **2.Reviewer's Comment**: "Additionally, though the authors show GPT-4 benefits from ToG compared to non-KG prompting (e.g., CoT), stronger prompting methods targeting for compositionality may be investigated, for example, self-ask [1]. Web search and vanilla retrieval augmentation methods can also be investigated [2]."
>
> **Reply**: We thank the reviewer's suggestion for comparing ToG with additional prompting-based method such as self-ask and with search and retrieval augmentation methods.
> |Prompting Methods w. ChatGPT | Accuracy on WebQSP|
> |--- |---|
> |CoT |62.2|
> |Self-ask|50.8|
> |Retrievel Augmentation with Wikipage| 61.6|
> |ToG|76.2|
> |ToG-R|75.8|
> Due to limited time of rebuttal, we only finish the experiment of Self-ask and Retrieval Augmentation with Wikipage on WebQSP dataset. We notice that the accuracy of self-ask method is 50.8%, and the accuracy of Retrieval Augmentation with Wikipage is 61.6%. The above experimental results are based on GPT-3.5-turbo which we use OpenAI API to call.  We would like to specifically note that the GPT3.5 turbo was updated recently. The experimental results of self-ask and RAG w. Wikipage are based on the latest GPT3.5 turbo model, while the other experimental results are based on the previous version of the GPT3.5 turbo model. We would like to evaluate Self-ask and RAG with Wikipage on all datasets used in this paper and update Table 1 with these results in the later version.
>
> **3.Reviewer's Comment**: "Efficiency of the method. The process of beam search and pruning with LLMs can be costly. An extensive comparison on decoding time and cost across all methods should be performed and discussed.
>
> **Reply**: Please refer to the official comment we submitted, named "**Response to all reviewers about the concerns on the efficiency of ToG**".

---

> ### Author Response · Authors · 2023-11-23
> **Kind Request for Timely Feedback on our Responses**
>
> Dear Reviewer,
>
> We are writing to follow up on the response we submitted regarding your comments.
>
> Firstly, we would like to express my gratitude for your initial review and valuable feedback. The insights provided have been instrumental in refining our paper, and we have tried our best to address each point in our response with the aim of enhancing the paper's quality and relevance.
>
> Understanding the time and effort required for a thorough review, we greatly appreciate the commitment you have shown towards ensuring the high standards of  ICLR. However, as the rebuttal process is concluding, we are keenly awaiting your further feedback on our responses. The comments from your expertise are crucial for the final preparation and improvement of our paper.
>
> If our responses more closely meet your expectations for the paper, we respectfully ask you to reconsider your initial rating.
> If you have any further questions or require more information to raise your initial score, please feel free to let us know. We are fully committed to making all necessary adjustments to meet your expectation.
>
> Thank you once again for your time and dedication. We look forward to your valuable feedback.

---

### Author Response · Authors · 2023-11-19
**Response to all reviewers about the concerns on the efficiency of ToG**

We agree with all reviewers that algorithmic efficiency is a very important issue for think-on-graph algorithm, and we thank reviewers for leading us rethink the efficiency problem. During the rebuttal period, we come up with new ideas to reduce the computational complexity of Think-on-Graph (proportional to the number of calling LLMs) **from the original $O(ND)$ to $O(D)$**, where $D$ is the depth (or equivalently length) of the reasoning path, and $N$ is the width of the beam-search (how many paths are remained in the pool in each iteration). Since we also have made other efforts for efficiency, we will state the details altogether in the following paragraphs, where we elaborate on the efficiency of original ToG in this official comment and point out how to further improve the efficiency of ToG as well.

**Improving ToG's Efficiency**

Before delving into the details, we would like to highlight the optimizations we have implemented to enhance the efficiency of ToG. effort we have made to improve ToG's computational efficiency.

●Solution 1: Reducing computational complexity from $O(ND)$ to $O(D)$ by using lightweight model in pruning.

●Solution 2: Reducing computational complexity from $O(ND)$ to $O(D)$ by unifying the prompts in the same pruning step.

●Solution 3: Optimizing pruning step to make the actual calls of LLMs much less than the previously estimated $2ND+D+1$ and closer to some common prompting methods such as CoT-SC.

**Detailed introduction of solutions**

**Details of Solution 1**: The bottleneck of computation is the pruning step, which contributes to $N*D$ times calling, and it is important to optimize it for computational efficiency. A technical route is to replace LLM with small models such as BM25 and Sentence-BERT in the pruning step since the small models are much faster than LLM calling.  In this way, we can reduce the number of LLM calling from $2ND+D+1$ to $D+1$. When $D$=3, for example, there are only 4 times LLM calling. However, this optimization sacrifices the accuracy due to the weaker scoring model in pruning. For example, as shown in Table 5 of the manuscript, the performance of ToG on WebQSP drops from 76.2% to 66.3% after replacing ChatGPT with SentenceBERT for pruning. To alleviate the issue of the performance degradation, we can appropriately increase the search width to compensate the loss because increasing search width can improve the chance of the optimal path to be selected in the pool and it doesn't affect the number of LLM calling. To empirically verify this, we increase the search width from 3 to 5 and reevaluate ToG with SentenceBERT as the pruning model on WebQSP. The accuracy rises to from 66.3% to 68.5% and could be further improved with a greater width since the greater width would not cause an increase in the number of LLM calls. We will highlight this point in the later version of our manuscript.

**Details of Solution 2**: Another solution on speeding up the pruning step is to employ the LLM at once to score all components of N candidate sets for obtaining top-N candidates, instead of calling the LLM N times to score N candidate sets separately. Through this solution, either entity pruning step or relation pruning step only need 1 LLM call for each iteration. Thus, the maximum number of LLM calls per question needed for ToG and ToG-R would drop to $2D+D+1$ and $D+D+1$. We are working on additional experiments based on this new method design and will update this comment as soon as new experimental  results are obtained. We apologize for not being able to update the new experimental results in time due to limted time for rebuttal. This is mainly due to the fact that the OpenAI API has been very unstable recently, which has significantly slow down our  experimental progress.

**Other Discussion About Efficiency**

We has discussed above the inference efficiency of ToG as well as two solutions for improvement. We would emphasize that, as a completely training-free method, ToG actually saves a lot of computational time training models (compared to pre-training or fine tuning methods). We hope the above explanations clarify the efficiency problem of ToG. We are committed to addressing the reviewers' concerns and continuously improving our approach. Thank you for the insightful feedback, which has helped us identify areas for optimization and refinement.

---

> ### Author Response · Authors · 2023-11-19
> **The original complexity (the number of LLM calls per question) of ToG**
>
> **Original complexity of ToG**
>
> For ToG and other LLM-based methods, the computational time (cost or complexity) in the inference phase mainly depends on how many times calling LLM.  For each question, ToG needs at most $2ND + D + 1$ times. Meanwhile, ToG-R needs at most $ND+D+1$ times as mentioned in Section 2.1.3 and 2.1.4. The details are introduced in the following paragraph.
>
> Given the beam search width $N$ and maximal reasoning depth $D$, ToG's initialize the search from the entity mostly aligning with the keyword in question. In each iterative step of the reasoning path, ToG starts from each of the $N$ entities/relations (nodes/edges on knowledge graph) and searches all its neighboring relations/entities. Given the search width $N$, ToG always keep $N$ "most-likely" candidate reasoning paths in the pool, and thus there are always $N$ candidate entity sets $E^D_{cand,n}$ and $N$ candidate relation sets $R^D_{cand,n}$. Consequently, it needs $N$ LLM calls for entity pruning and $N$ calls for relation pruning, respectively, as well as one additional LLM call for reasoning (evaluating if the information from the current candidate paths are enough or not).  We have to point it out that, for each of the $N$ starting entities, all its neighbor entities/relations are NOT scored one by one. On the contrary, all its neighbor entities/relations are "translated" into "one" prompt altogether and are sent to LLM, which output the top-$N$ candidates at one-time. Therefore, each starting entity only calls LLM once for pruning and so $N$ starting entities calls LLM $N$ times in one iterative step. Consequently, there are totally $2ND+D$ times calling after reasoning $D$ steps. In the end, there is an additional calling that "translate" the final path to user-understandable language and answer the user.  Therefore, ToG requires $2ND+D+1$ LLM calls in total. Since most questions can be answered within 3 hops (means depth of reasoning path is 3), and the performance is usually good enough when the search width $N$=3 as we tested in Figure 3, the total number of LLM calling is $2\times 3\times 3+3+1=22$. So the computational time is about 21 times longer than that of LLM-only.
>
> **Note 1**: With a similar performance to ToG, its variant ToG-R only calls LLM for $ND+D+1$ times by using random entity pruning instead of LLM-based entity pruning, saving nearly half of computational time.
>
> **Note 2**: $2ND+D+1$ is the maximal computational complexity. In most cases, ToG does not need $2ND+D+1$ LLM calls for a question because the whole reasoning process might be early stopped before the maximum reasoning depth D is reached if LLM determines enough information has been retrieved. Likewise, ToG-R does not really need $ND+D+1$ LLM calls in most cases. As an illustration, the following table shows the AVERAGE numbers of LLM calls per question needed by ToG on different datasets. It can be seen that in the four multi-hop KBQA datasets, the average numbers of LLM calls (ranging from 10 to 15) are significantly smaller than 22, which is the theoretical maximum number of LLM calls calculated from $2ND+D+1$ when $N$=3 and $D$=3. We can also see that this AVERAGE number gets even smaller (< 10) for single-hop reasoning datasets, such as SimpleQuestion and T-REx.
>
>
> | |  CWQ|WebQSP|GrailQA|QALD10-EN|SimpleQuestion|WebQuestion|T-REx|Zero-Shot RE|Creak|
> |---|  ------|------|------|------|------|------|------|------|------|
> | The average number of LLM calls per question|  14.3	|11.2|10.4|11.4|8.7|10.5|7.7|7.6|8|

---

### Public Comment · ~Yu_Gu5 · 2023-12-02
**public comment by Yu Gu from OSU NLP**

Hi, I am the author of Pangu [1].

I recently came across this paper and found it interesting. Please accept my apologies for being this late to join the discussion. Actually, I only have one smaller point of clarification to respectfully raise for consideration by the authors.

This paper claims "we propose a new tight-coupling LLM ⊗ KG paradigm where KGs and LLMs work in tandem, complementing each other’s capabilities in each step of graph reasoning", motivated by the statement that existing work follows the LLM⨁KG paradigm which suffers from loose coupling. We believe this is a wonderful insight! However, we want to note that Pangu [1] also leverages LLMs to reason over the KG step by step, in a very similar manner. For a clear visualization of Pangu on KG, feel free to check the demo made by us: https://twitter.com/ysu_nlp/status/1605233874077876225

The paper, while sharing a similar overarching spirit, appears to have inadvertently overlooked a discussion on this particular aspect.
I would greatly appreciate if the authors could add more discussion comparing with Pangu in the camera-ready version. Here, I briefly outline the two key differences between ToG and Pangu, if that's helpful.

a. In the LLM setting, Pangu uses the logprobs of the LLM as a scoring function to rank candidates, while ToG directly ask the LLM to output the score for candidate, but different design choices can be easily swapped here. This is more of an implementation-level thing.

b. For each reasoning step over the KG, Pangu operates at the sub-program level while ToG operates at the relation-entity level. The purpose of Pangu is to reason over the KG to find the proper program for deterministic execution. By contrast, ToG reasons over the KG to find useful triples to assist the LLM in the final answer prediction. The divergence in approach may stem from the differing aims of the two works - Pangu targets grounded language understanding, whereas ToG focuses on responsible text generation.


To make it clear to the reviewers & AC, I think this paper is an interesting paper that makes a compelling case for tightly coupling LLMs with the KG to facilitate responsible generation, especially for question answering. **My aim is solely to provide additional comment, not negative critique.**


[1] Don’t Generate, Discriminate: A Proposal for Grounding Language Models to Real-World Environments. https://arxiv.org/abs/2212.09736

---

### Meta-Review · Area_Chair_1RrY · 2023-12-09

**Metareview:**

This paper studies the problem of integrating knowledge graphs (KGs) in LLMs. It proposes the Think-on-Graph (ToG) method that performs beam-search on KGs. ToG uses a "search" operation backed by the KG and a "prune" operation backed by the LLM for exploring the reasoning paths. On five KBQA datasets with extensive experiments, the paper justifies the effectiveness of the proposed ToG method. The analyses on each component of the method is extensive. The reviewers have concerns on the efficiency of the approach, as it requires more LLM calls during reasoning. Also, the performance of the proposed ToG seems rely heavily on the use of very strong LLMs (e.g., the performance advantage on 6 out of 9 benchmarks is only observed with GPT-4). It's desirable to have results and analyses on small LLMs such as 7B and 13B.

**Justification For Why Not Higher Score:**

- Concerns on the efficiency of the approach, as it requires more LLM calls during reasoning.
- More results on small LLMs are desirable. The performance of the proposed ToG seems rely heavily on the use of very strong LLMs (e.g., GPT-4).

**Justification For Why Not Lower Score:**

- Intuitive new method
- Extensive experiments showing improvement

---

### Decision · Program_Chairs · 2024-01-16

Accept (poster)